# Beyond scalar losses: calibrating segmentation models via gradient vector field surgery

**Laurin Lux**[1,2,3]                                                                    LAURIN.LUX@TUM.DE

**Alexander H. Berger**[1,3,4]                                                           A.BERGER@TUM.DE

**Moritz Knolle**[3]                                                                    MORITZ.KNOLLE@TUM.DE

**Daniel Rückert**[1,2,4,6]                                                             DANIEL.RUECKERT@TUM.DE

**Johannes C. Paetzold**[4,5]                                                     JPAETZOLD@MED.CORNELL.EDU

[1] *School of Computation, Information and Technology, TUM, Munich, Germany*

[2] *Munich Center for Machine Learning, Munich, Germany*

[3] *Department of Radiology, Weill Cornell Medicine, New York City, USA*

[4] *School of Medicine and Health, TUM University Hospital, Munich, Germany*

[5] *Cornell Tech, New York City, USA*

[6] *Department of Computing, Imperial College London, London, UK*

**Editors:** Accepted for publication at MIDL 2026

## Abstract

Region-based loss functions, such as the Dice loss, have established themselves as the de facto standard for highly class- and region-imbalanced segmentation tasks. However, models trained using region-based loss functions are notoriously miscalibrated and typically yield over-confident predictions. In medical imaging applications, such as defining tumor resection margins, this miscalibration is hindering clinical adoption. In this work, we outline a novel gradient perspective on this overconfidence and show how it affects region-based loss functions. We propose a "surgery" on the gradient vector field as a simple, yet effective intervention to mitigate calibration issues. This surgery adds a factor to the loss's partial derivative, scaling the gradient's magnitude linearly with the prediction error. In empirical evaluations across 2D and 3D medical segmentation tasks, we demonstrate the effectiveness of this intervention while maintaining high prediction accuracy when used in conjunction with any region-based loss function.

**Keywords:** Segmentation, Calibration, Optimization, Gradient Surgery, Metastases

## 1. Introduction

The Dice Similarity Coefficient (DSC) has become a primary evaluation metric and loss function in medical image segmentation. Originally adapted for volumetric segmentation (Milletari et al. (2016)), the Dice loss and its derivatives (e.g. (Salehi et al., 2017; Taghanaki et al., 2019) excel in scenarios of extreme class imbalance—a ubiquitous challenge in medical imaging where foreground structures (e.g., lesions or vessel fragments) occupy negligible fractions of the image volume. By directly optimizing a continuous approximation of the region overlap between predictions and ground truth, the Dice loss circumvents the local minima often encountered when training voxel-wise objectives on highly imbalanced data.

However, this robustness to class imbalance comes at a cost. The Dice loss cannot inherently enforce probabilistic consistency with the underlying data-generation process,

unlike e.g. the Cross-Entropy (CE) loss, which corresponds directly to a proper scoring rule (Gneiting and Raftery, 2007). Instead, models trained with Dice loss exhibit pathological overconfidence, pushing softmax probabilities toward 0 or 1 regardless of the actual epistemic uncertainty. This creates a significant dichotomy for clinical model development. In high-stakes workflows, such as defining tumor resection margins or radiotherapy target volumes, a segmentation map is not merely a binary mask but a decision boundary. Well-calibrated predictions enable meaningful verification and the imperative possibility of adapting outputs to high-recall or high-sensitivity solutions (Sander et al., 2019; Jiang et al., 2012).

In this work, we analyze partial derivatives w.r.t. the logits, influencing the gradient on the network's weights, to identify the root cause of miscalibration inherent to all region-based segmentation losses. We show that the gradient dynamics of these losses effectively neglect the calibration of the predicted probabilities and only optimize for region overlap between predictions and ground truth. To mitigate this issue, we propose a *gradient surgery*, a simple yet effective intervention (surgery) on the gradient vector field of the model's voxel-wise logit outputs. Given a network's predicted probability $p$, this intervention rescales the loss's partial derivative w.r.t. single pixel logits such that the error $|y - p|$ has a linear influence. In extensive empirical experiments on 2D and 3D medical segmentation tasks, we show that our proposed method improves model calibration while maintaining high segmentation performance.

## 2. Related work

Seminal works by Mehrtash et al. (2020); Bertels et al. (2019); Sander et al. (2019) demonstrate that segmentation models trained with Dice loss provide miscalibrated, overconfident predictions and thus questioned their clinical applicability. Initial mitigation strategies included model ensembles to improve the calibration of such region-based losses (Mehrtash et al., 2020). Other common strategies involve compound objectives, such as the Combo Loss (Taghanaki et al., 2019) or Unified Focal Loss (Yeung et al., 2022), which compute a weighted sum of Dice and CE (or Focal) terms. While these stabilize training, they often require extensive tuning of the weighting hyperparameter $\lambda$ and represent a compromise rather than a theoretical fix for the miscalibration. The marginal L1 average calibration error (mL1-ACE) was recently proposed as an auxiliary loss that is specifically targeted at improving voxel-wise calibration (Barfoot et al., 2024). Another focus was the direct adaptation of region-based losses. The Tversky Loss (Salehi et al., 2017) generalizes the Dice coefficient to allow for individual weighting of false positives and false negatives, which impacts precision and recall but does not explicitly address probabilistic calibration. More recently, DSC++ (Yeung et al., 2023) introduced an exponent $\gamma > 1$ to the Dice formulation to selectively penalize overconfident, incorrect predictions. While the focal $\gamma$ results in improved calibration, it can drastically change the gradient dynamics compared to the Dice loss through the down-weighting of samples with large fractions of false positives and false negatives (see Appendix E). An alternative to modifying the primary loss is post-hoc recalibration (Rousseau et al., 2021). Techniques such as temperature scaling (Guo et al., 2017), Platt scaling (Platt et al., 1999), and isotonic regression map (Zadrozny and Elkan, 2002) model outputs to calibrated probabilities after training. While effective on in-distribution

data, these methods do not improve the quality of the learned feature representation and are known to degrade under the domain shifts common in medical deployment.

Other works (Islam and Glocker, 2021; Murugesan et al., 2025, 2023a; Karani et al., 2023) have focused on specific solutions for the uncertainty specific to lesion boundaries that are inherent to the data annotation process. Spatially varying labels smoothing (SVLS) (Islam and Glocker, 2021) draws inspiration from label smoothing, specifically smoothing the voxels with varying neighbor annotations (i.e., boundary voxels). This method improves calibration for brain tumor, kidney tumor, long nodule, and prostate zone segmentation. Neighbor-Aware Calibration (NACL) (Murugesan et al., 2025) reformulates and extends SVLC by treating it as a neighborhood-aware penalty. Moreover, it applies a constraint directly on the logits (Liu et al., 2022), effectively reducing their magnitude. The penalty formulation allows flexible weighting of the initial optimization objective with the neighborhood-aware logit distance constraint. Finally, boundary-weighted consistency regularization (BWCR) (Karani et al., 2023) forces logit consistency across corresponding pixels from different augmented versions of the same input. However, all of these methods are not specifically designed to address calibration issues in models trained with region-based losses. In contrast, our work specifically targets the overconfidence problem in region-based losses, aiming to improve performance when region-based losses are preferred over standard cross-entropy loss.

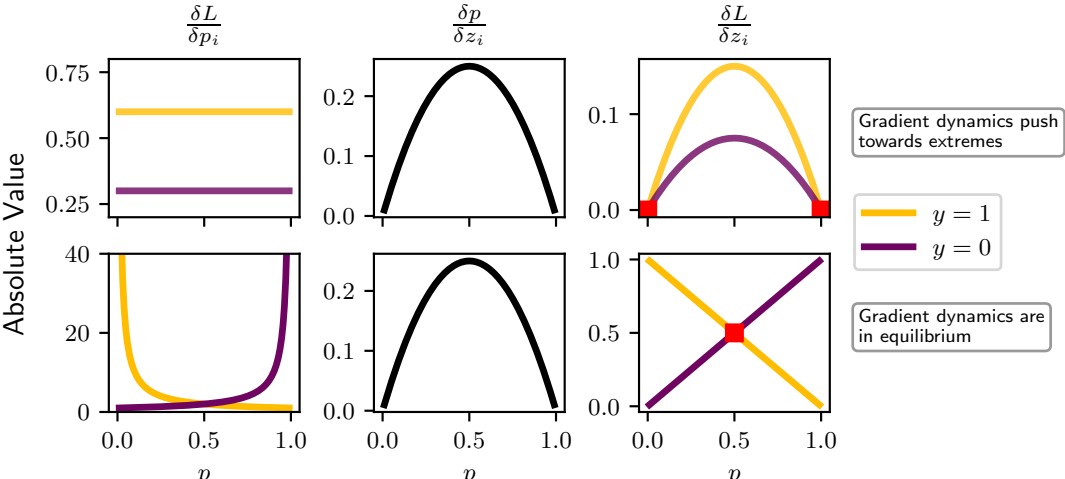

Figure 1: Partial derivatives of dice loss (top row) vs. cross entropy loss (bottom row) for single voxels. Sub-panels show the absolute value of: $\frac{\delta L}{\delta p_i}$, $\frac{\delta p}{\delta z_i}$, and $\frac{\delta L}{\delta z_i}$ as a function of the predicted probability $p$ for a foreground ($y = 1$, yellow) and a background ($y = 0$, purple) voxel. Red squares indicate intersection points where the magnitude of foreground and background derivatives is in equilibrium. For cross-entropy, the curves intersect at $p = 0.5$, encouraging uncertain predictions for indistinguishable voxel-representations with different labels. For the Dice loss, they intersect at $p \in \{0, 1\}$, effectively pushing *all* predicted probabilities to extreme values.

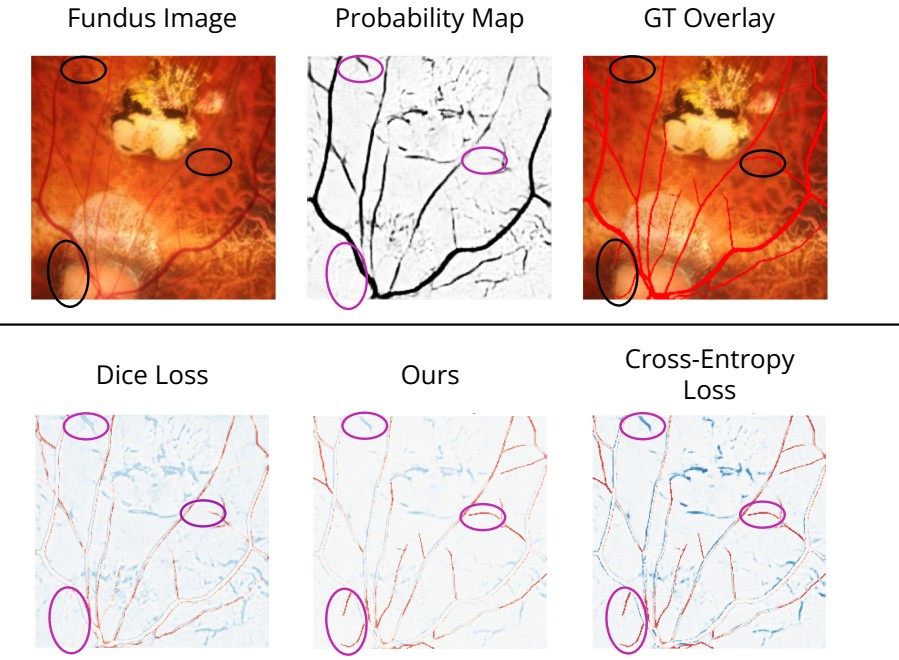

Gradient w.r.t. to pixel logits

Figure 2: Visualization of the gradient w.r.t. the voxel-wise logits. Purple circles indicate confident errors, where gradients vanish through the activation function for Dice.

## 3. Gradient dynamics of region-based segmentation losses: analysis and intervention

Below, we present a concise analysis of the Dice loss's gradient dynamics alongside our proposed intervention that encourages calibrated predictions. We assume a binary segmentation problem using a final sigmoid activation function to turn logits into probability values.

### 3.1. Region-based losses converge to miscalibrated solutions

The soft dice loss for a prediction/target mask pair $P \in \mathbb{R}^N$ and $Y \in \{0, 1\}^N$ is defined as:

$$\text{DSC}(P,Y) = 1 - \frac{2\sum_{i=1}^{N} p_i y_i + \epsilon}{\sum_{i=1}^{N} p_i + \sum_{i=1}^{N} y_i + \epsilon} = 1 - \frac{2I + \epsilon}{P + Y + \epsilon}, \quad (1)$$

where $I$ is the intersection between $P$ and $Y$. The partial derivative of the Dice loss w.r.t. the predicted probability $p_i$ for an individual input voxel $i$ is:

$$\frac{\partial L_{DSC}}{\partial p_i} = \frac{2y_i(P + Y + \varepsilon) - (2I + \varepsilon)}{(P + Y + \varepsilon)^2}, \quad (2)$$

which we will refer to as the "global" term $G(i)$. Crucially, as with all region-based losses, $G(i)$ can usually be approximated as a constant $G$ because a single voxel has negligible

influence on $G$ as the image size $N$ increases. Using sigmoid/softmax activation on the model's output, the partial derivative of the loss w.r.t. a single voxel logit $z_i$ is:

$$\frac{\partial L_{DSC}}{\partial z_i} = \frac{\partial L_{DSC}}{\partial p_i} \cdot \frac{\partial p_i}{\partial z_i} = G(i) \cdot p_i(1 - p_i). \tag{3}$$

These partial derivatives have two undesirable properties, visualized in Figure 1. First, $\partial L_{DSC}/\partial z_i$ is maximized by uncertain voxels ($p \approx 0.5$) while confident predictions ($p \approx 0$ or $p \approx 1$) have negligible influence on the gradient w.r.t. the network's parameters. Importantly, this occurs independently of their correctness; i.e., confident but incorrect predictions do not contribute to the gradient w.r.t. the network weights, as shown in Figure 2. Second, the partial derivatives for foreground and background intersect only at the function's boundaries, i.e., 0 and 1; see red squares in Figure 1, top row, which causes the network to converge to overconfident predictions. We make a simplified argument by considering a scenario where a network is trained to a point where it has exhausted its maximal discriminative capacities, i.e., there exist voxels $a$ and $b$ with different labels ($y_a = 1$ and $y_b = 0$) that are indistinguishable through the network's latent representation $\mathbf{l}(x)$, i.e. $\mathbf{l}(a) \approx \mathbf{l}(b)$. Therefore, the network is forced to output highly coupled probabilities for both voxels, i.e., $p_a \approx p_b$. The described scenario naturally evolves when a network is trained towards convergence without reaching zero loss. In this scenario, $a$ and $b$ influence the gradient w.r.t. the network parameters in opposite directions through the opposing ground truth labels for these "indistinguishable" voxels. For $y_a = 1$, the network parameters are guided towards a higher probability, and for $y_b = 0$ towards a lower probability. Therefore, the network's weights converge to output the probability for $\mathbf{l}(a)$ and $\mathbf{l}(b)$ such that the influence of $a$ and $b$ on the gradient w.r.t. the network weights is in an equilibrium (red squares in Figure 1):

$$\frac{\partial L}{\partial z_a} = -\frac{\partial L}{\partial z_b} \tag{4}$$

Formally, this equilibrium can be extended from a single pair of indistinguishable voxels $v$ to sets of indistinguishable voxels $S_k := \{v \mid l(v) \approx c_k\}$, where $c_k$ is the voxel set's shared latent representation. As described above, the network predicts the same $p_k$ for all elements (voxels) in a set $S_k$. In addition to $p_k$, a set $S_k$ is characterized by its ratio between foreground and background labels $r_k = |S_k^{fg}|/|S_k|$, where $S_k^{fg} := \{v \in S_k | y(v) = 1\} \subset S_k$ is the subset of $S_k$ containing the voxels with ground truth label $y = 1$. These sets are naturally encountered when training on complete samples/batches consisting of a large number of voxels, whose influences accumulate, resulting in numerous different equilibrium probabilities that characterize the network's calibration.

In the case of cross-entropy, the equilibrium for any set of indistinguishable voxels $S_k$ with foreground ratio $r_k$ is reached at $p_k = r_k = |S_1|/|S|$. Note that this directly corresponds to perfect calibration, where the predicted probabilities correspond to the underlying data-generating distribution. (Guo et al., 2017).

In the case of Dice, an equilibrium can only be reached for $p_a = p_b = 0 \vee p_a = p_b = 1$, which is independent of the set's label ratio and leads to overconfident predictions that are unrelated to the underlying data-generating distribution.

### 3.2. Combining calibration and region size imbalance awareness using gradient surgery

We hypothesize that ideally, the partial derivatives w.r.t. the voxel logits respect (1) error magnitude to obtain equilibria resulting in calibrated probability outputs (similar to CE, see Figure 1), and (2) dynamic adaptation to drastic region size imbalance for overlap maximization. Suitable partial derivatives for foreground and background that fulfill these requirements are:

$$\frac{\partial L}{\partial z_i^{fg}} = (1 - p_i)\frac{2(P + Y + \varepsilon) - (2I + \varepsilon)}{(P + Y + \varepsilon)^2} = (1 - p_i)\, G^{fg}(i), \tag{5}$$

$$\frac{\partial L}{\partial z_i^{bg}} = -p_i\frac{(2I + \varepsilon)}{(P + Y + \varepsilon)^2} = -p_i\, G^{bg}(i), \tag{6}$$

respectively. Here, the magnitude scales linearly with the error while maintaining adaptive, region-size-dependent foreground and background weighting. Notably, the global terms $G^{fg}$ and $G^{bg}$ are equal to the Dice formulation. Following the chain rule, where the partial derivative of the probability w.r.t. the logits is $\partial p_i/\partial z_i = (1 - p_i)*p_i$, we would need a scalar loss that results in the following partial derivatives w.r.t. the single voxel probabilities:

$$\frac{\partial L}{\partial p_i^{fg}} = \frac{1}{p_i}\frac{2(P + Y + \varepsilon) - (2I + \varepsilon)}{(P + Y + \varepsilon)^2} = \frac{1}{p_i}G^{fg}(i), \tag{7}$$

$$\frac{\partial L}{\partial p_i^{bg}} = -\frac{1}{(1 - p_i)}\frac{(2I + \varepsilon)}{(P + Y + \varepsilon)^2} = -\frac{1}{(1 - p_i)}G^{bg}(i), \tag{8}$$

For these partial derivatives to form the gradient w.r.t. the logits $\nabla_{\mathbf{z}}L$ for a scalar loss function $L$, we require symmetry of second derivatives, which is not guaranteed for all $z_i$ and $z_k$ as outlined in the proof in Appendix A. We identify this as the reason no loss with the desired partials was previously proposed. Instead of relying on a scalar loss, we define a vector field $\mathcal{F}(\mathbf{z})$ with

$$\mathcal{F}_i(\mathbf{z}) = \frac{p_i(1 - p_i)}{(y_i p_i + (1 - y_i)(1 - p_i))}\frac{2y_i(P + Y + \varepsilon) - (2I + \varepsilon)}{(P + Y + \varepsilon)^2}, \tag{9}$$

that we use as an optimization objective for model training. Our gradient scale factor is $p_i$ when $y_i = 0$ and $1 - p_i$ when $y_i = 1$. This can be interpreted as either a "region-imbalance" weighted cross-entropy gradient, or, as a linearly error-weighted dice gradient without the sigmoid derivative. Implementation details of the methods are described in Appendix D.

Empirically, we find that adding a relatively sharp decline near 0 and 1 results in higher performance. We add this sharp decline by multiplying by $(1 - (1 - p)^n)$ and $(1 - p^n)$, where $n$ regulates the steepness of the decline. For $n \to \infty$ the function is essentially equivalent to $|y - p|$ for $0 < p < 1$, and 0 for $p \in \{0, 1\}$. Effectively, for $|y - p|$ values close to 0, this has similarity to label smoothing for cross-entropy loss (Szegedy et al., 2016; Müller et al., 2019), by reducing the incentive of the model to push probabilities to maximal certainty. Symmetrically, for $|y - p|$ close to 1, this can be interpreted as de-emphasizing extremely confident errors, potentially improving robustness against obvious cases of label noise. An

ablation on the exponential $n$ is displayed in section 4.2. Including the decline terms, the vector field is defined as:

$$\mathcal{F}_i(\mathbf{z}) = (1 - (1 - p)^n)(1 - p^n)\frac{p_i(1 - p_i)}{(y_i p_i + (1 - y_i)(1 - p_i))}\frac{2y_i(P + Y + \varepsilon) - (2I + \varepsilon)}{(P + Y + \varepsilon)^2}, \quad (10)$$

### 3.3. Vector field stability

The non-existence of a scalar loss function yielding the desired partial derivatives implies that the logit vector field and, therefore, also the induced vector field on the network weights, is non-conservative. A non-zero curl of vector fields can result in difficulties during optimization that are well-studied, e.g., in the field of generative adversarial networks (Mescheder et al., 2017). However, the curl in our proposed vector field $\mathcal{F}(\mathbf{z})$ is negligible compared to the diagonal terms, which prevents the problematic "orbiting" around solutions. A visualization of $\mathcal{F}(\mathbf{z})$ compared to the gradient vector fields of other methods is displayed in Figure 4 in Appendix B. Moreover, $\mathcal{F}(\mathbf{z})$ provides favorable theoretical properties that make it suitable for model training: First, the proposed vector field $\mathcal{F}(\mathbf{z})$ is continuous and smooth on $\mathbb{R}^\varkappa$, since each component $\mathcal{F}_i(\mathbf{z})$ is a smooth function, see Equation 10. Second, the vector field always points towards the ground truth solution $\mathbf{g}$, since no sign flips occur for the components $\mathcal{F}_i(\mathbf{z})$. These theoretical considerations, in conjunction with our empirical evaluation in Section 4, showcase the proposed solution's suitability for effective network training. Example training curves with different optimizers are displayed in Appendix C.

## 4. Experimentation and Results

We compare our custom vector field adaptations for different region-based losses, including Dice, Tversky (Salehi et al., 2017), Combo loss (CE + Dice) (Taghanaki et al., 2019), m1L1-ACE (+Dice) loss, and Dice++ losses. Moreover, we include baselines employing spatially aware label smoothing (SVLS) (Islam and Glocker, 2021) and neighbor-aware calibration through penalty constraints (NACL (Murugesan et al., 2025)). Notably, these were not designed to work in conjunction with region-based losses (Murugesan et al., 2023b). Details on the hyperparameter settings for these losses are listed in Appendix C. We conduct a random hyperparameter search to find the optimal configuration for each setup with 25 and 10 runs for our 2D and 3D datasets, respectively. Implementation details for our optimization method are provided in the Appendix D. We use the UNet architecture (Ronneberger et al., 2015) with residual units (He et al., 2016) combined with heavy domain-specific augmentations (Isensee et al., 2021).

**Metrics** We evaluate model calibration using negative log-likelihood (NLL), expected calibration error (ECE), maximum calibration error (MCE) (Naeini et al., 2015; Guo et al., 2017), and Brier score (Glenn et al., 1950). Calibration metrics are calculated on all voxels; a comparison to a calculation on the "active" foreground region defined as the union of target and prediction foreground is displayed in Appendix H. Additionally, we report the Dice similarity coefficient (DSC) as an overlap-based metric.

Table 1: Average test set performance of the 5 best runs out of 25 (selected through validation Dice scores) trained using each loss on the INbreast dataset (Moreira et al., 2012). Best results are displayed in **bold**, second-best results are underlined. Significantly better performance for standard losses vs. altered losses is highlighted through a $^*$ using the 0.01 significance level.

|  | Method | NLL ↓ | ECE ↓ | MCE ↓ | Brier ↓ | DSC (%) ↑ |
|---|---|---|---|---|---|---|
| INbreast Masses | Cross Entropy | $0.0192_{\pm 0.0035}$ | $0.0038_{\pm 0.0015}$ | $0.1903_{\pm 0.0872}$ | $0.0050_{\pm 0.0008}$ | $66.21_{\pm 3.57}$ |
| | NACL | $0.0208_{\pm 0.0044}$ | $0.0038_{\pm 0.0010}$ | $0.2390_{\pm 0.1021}$ | $0.0050_{\pm 0.0008}$ | $68.22_{\pm 3.57}$ |
| | NACL + Dice | $0.0269_{\pm 0.0075}$ | $0.0040_{\pm 0.0007}$ | $0.3185_{\pm 0.0121}$ | $0.0045_{\pm 0.0007}$ | $72.44_{\pm 2.62}$ |
| | SVLS | $0.0178_{\pm 0.0027}$ | $0.0035_{\pm 0.0010}$ | $0.2161_{\pm 0.0816}$ | $0.0045_{\pm 0.0006}$ | $68.35_{\pm 2.90}$ |
| | SVLS + Dice | $0.0277_{\pm 0.0049}$ | $0.0045_{\pm 0.0008}$ | $0.2925_{\pm 0.0138}$ | $0.0052_{\pm 0.0008}$ | $68.76_{\pm 2.70}$ |
| | Dice++ | $0.0195_{\pm 0.0024}$ | $0.0034_{\pm 0.0009}$ | $0.1774_{\pm 0.0244}$ | $0.0045_{\pm 0.0007}$ | $67.51_{\pm 3.17}$ |
| | CE + Dice | $0.0328_{\pm 0.0067}$ | $0.0049_{\pm 0.0007}$ | $0.3111_{\pm 0.0130}$ | $0.0054_{\pm 0.0006}$ | $71.76_{\pm 2.10}$ |
| | CE + Dice - Surgery | $\mathbf{0.0153}_{\pm 0.0013}$ | $\mathbf{0.0023}_{\pm 0.0003}$ | $\underline{\mathit{0.1568}}_{\pm 0.0092}$ | $\mathbf{0.0039}_{\pm 0.0003}$ | $\mathbf{74.39}_{\pm 3.34}$ |
| | Dice | $0.0567_{\pm 0.0160}$ | $0.0058_{\pm 0.0006}$ | $0.3214_{\pm 0.0324}$ | $0.0062_{\pm 0.0005}$ | $64.93_{\pm 3.31}$ |
| | Dice - Surgery | $\underline{0.0164}^*_{\pm 0.0042}$ | $\underline{0.0026}^*_{\pm 0.0010}$ | $\mathbf{0.1495}^*_{\pm 0.0112}$ | $\underline{0.0039}^*_{\pm 0.0009}$ | $\underline{74.34}^*_{\pm 2.77}$ |
| | Tversky | $0.0489_{\pm 0.0114}$ | $0.0051_{\pm 0.0012}$ | $0.3439_{\pm 0.0372}$ | $0.0054_{\pm 0.0012}$ | $67.63_{\pm 2.90}$ |
| | Tversky - Surgery | $0.0187^*_{\pm 0.0035}$ | $0.0030_{\pm 0.0008}$ | $0.1650^*_{\pm 0.0184}$ | $0.0047_{\pm 0.0009}$ | $69.34_{\pm 3.76}$ |
| FIVES | Cross Entropy | $0.0542_{\pm 0.0011}$ | $\mathbf{0.0043}_{\pm 0.0001}$ | $\mathbf{0.0467}_{\pm 0.0015}$ | $0.0138_{\pm 0.0002}$ | $87.54_{\pm 0.22}$ |
| | NACL | $0.0536_{\pm 0.0005}$ | $\mathbf{0.0043}_{\pm 0.0000}$ | $\underline{0.0478}_{\pm 0.0015}$ | $0.0138_{\pm 0.0001}$ | $87.60_{\pm 0.14}$ |
| | NACL + Dice | $0.0659_{\pm 0.0023}$ | $0.0103_{\pm 0.0005}$ | $0.1548_{\pm 0.0105}$ | $0.0143_{\pm 0.0002}$ | $87.94_{\pm 0.11}$ |
| | Dice++ | $0.0545_{\pm 0.0006}$ | $0.0048_{\pm 0.0002}$ | $0.0560_{\pm 0.0031}$ | $\mathbf{0.0135}_{\pm 0.0001}$ | $87.97_{\pm 0.07}$ |
| | Dice + m1L1-ACE | $0.0756_{\pm 0.0020}$ | $0.0074_{\pm 0.0003}$ | $0.0613_{\pm 0.0054}$ | $0.0156_{\pm 0.0004}$ | $85.96_{\pm 0.45}$ |
| | CE + Dice | $0.0662_{\pm 0.0013}$ | $0.0103_{\pm 0.0002}$ | $0.1558_{\pm 0.0047}$ | $0.0142_{\pm 0.0002}$ | $\underline{87.99}_{\pm 0.16}$ |
| | CE + Dice - Surgery | $\underline{0.0533}^*_{\pm 0.0014}$ | $0.0045^*_{\pm 0.0003}$ | $0.0486^*_{\pm 0.0024}$ | $\underline{0.0136}^*_{\pm 0.0001}$ | $87.70_{\pm 0.09}$ |
| | Dice | $0.3118_{\pm 0.0864}$ | $0.0162_{\pm 0.0006}$ | $0.3514_{\pm 0.0144}$ | $0.0165_{\pm 0.0005}$ | $\mathbf{88.03}_{\pm 0.25}$ |
| | Dice - Surgery | $0.0540^*_{\pm 0.0013}$ | $0.0044^*_{\pm 0.0003}$ | $0.0485^*_{\pm 0.0032}$ | $0.0139^*_{\pm 0.0001}$ | $87.60_{\pm 0.12}$ |
| | Tversky | $0.2595_{\pm 0.0680}$ | $0.0163_{\pm 0.0001}$ | $0.3408_{\pm 0.0127}$ | $0.0168_{\pm 0.0002}$ | $87.77_{\pm 0.21}$ |
| | Tversky - Surgery | $\mathbf{0.0532}^*_{\pm 0.0009}$ | $\underline{0.0044}^*_{\pm 0.0002}$ | $0.0488^*_{\pm 0.0018}$ | $\underline{0.0136}^*_{\pm 0.0002}$ | $87.79_{\pm 0.25}$ |

**Datasets**  We perform experiments on datasets for 2D retinal vessel segmentation on the FIVES dataset (Jin et al., 2022), for 2D mass segmentation in mammography images (Moreira et al., 2012), for 3D metastasis segmentation on the BraTS-METS dataset (Maleki et al., 2025), and 3D tumor segmentation on the KiTS dataset (Heller et al., 2019). Detailed descriptions of the datasets and data splits are provided in Appendix C.

Table 2: Best out of 10 runs (selected through validation Dice scores) trained using each loss on the BraTS-METS dataset (Maleki et al., 2025) (top block) and the KiTS dataset (Heller et al., 2019) (bottom block). Best results are displayed in **bold**, second-best results are underlined. Better performance for standard losses vs. altered losses is highlighted through *italics*.

| | Method | NLL ↓ | ECE ↓ | MCE ↓ | Brier ↓ | DSC (%) ↑ |
|---|---|---|---|---|---|---|
| **BraTS Metastasis** | Dice++ | 0.0187 | 0.0006 | **0.1359** | 0.0017 | 72.53 |
| | CE + Dice | 0.0094 | 0.0007 | 0.3219 | 0.0016 | 73.05 |
| | CE + Dice - Surgery | **0.0051** | **0.0005** | *0.1451* | 0.0015 | **73.19** |
| | Dice | 0.0234 | 0.0009 | 0.4113 | 0.0019 | 69.53 |
| | Dice - Surgery | 0.0055 | **0.0005** | *0.1414* | **0.0014** | *71.02* |
| | Tversky | 0.0342 | 0.0010 | 0.4273 | 0.0019 | *72.39* |
| | Tversky - Surgery | *0.0265* | **0.0005** | *0.1379* | *0.0016* | 71.80 |
| **KiTS Metastasis** | CE | 0.0139 | 0.0019 | 0.1845 | 0.0058 | 64.79 |
| | SVLS | **0.0115** | 0.0020 | 0.1401 | 0.0059 | 70.68 |
| | SVLS + Dice | 0.0203 | 0.0024 | 0.2686 | 0.0056 | 75.82 |
| | NACL | 0.0614 | 0.0509 | 0.1640 | 0.0110 | 64.48 |
| | NACL + Dice | 0.0307 | 0.0176 | 0.2096 | 0.0075 | 74.56 |
| | Dice++ | 0.0165 | 0.0024 | 0.1459 | 0.0066 | 75.57 |
| | CE + Dice | 0.0238 | 0.0029 | 0.2884 | 0.0064 | 75.77 |
| | CE + Dice - Surgery | 0.0125 | **0.0017** | *0.1389* | **0.0052** | **76.75** |
| | Dice | 0.0580 | 0.0033 | 0.3513 | 0.0068 | *76.62* |
| | Dice - Surgery | *0.0161* | *0.0022* | **0.1345** | *0.0063* | 74.01 |
| | Tversky | 0.0714 | 0.0041 | 0.3616 | 0.0083 | 73.10 |
| | Tversky - Surgery | *0.0229* | *0.0029* | *0.1404* | *0.0077* | *73.87* |

## 4.1. Results

Tables 1 and 2 present our main results for the experiments on 2D and 3D datasets. Our proposed gradient vector field surgery, applied to the gradient of a region-based loss function, improves calibration metrics compared to the respective baseline losses alone (ComboLoss, Dice, and Tversky) across all cases in all datasets. In some cases, our approach reduces NLL and ECE by factors of 4 to 6, with negligible (FIVES, BraTS, KiTS) or positive (INbreast) impact on binary prediction performance. Furthermore, our approach applied to varying baseline losses consistently yields the best (INBreast, KiTS, BraTS) or second-best (FIVES, BraTS) calibration scores in all metrics. Especially CE + Dice with gradient vector field surgery performs strongly on all datasets in terms of calibration and DSC.

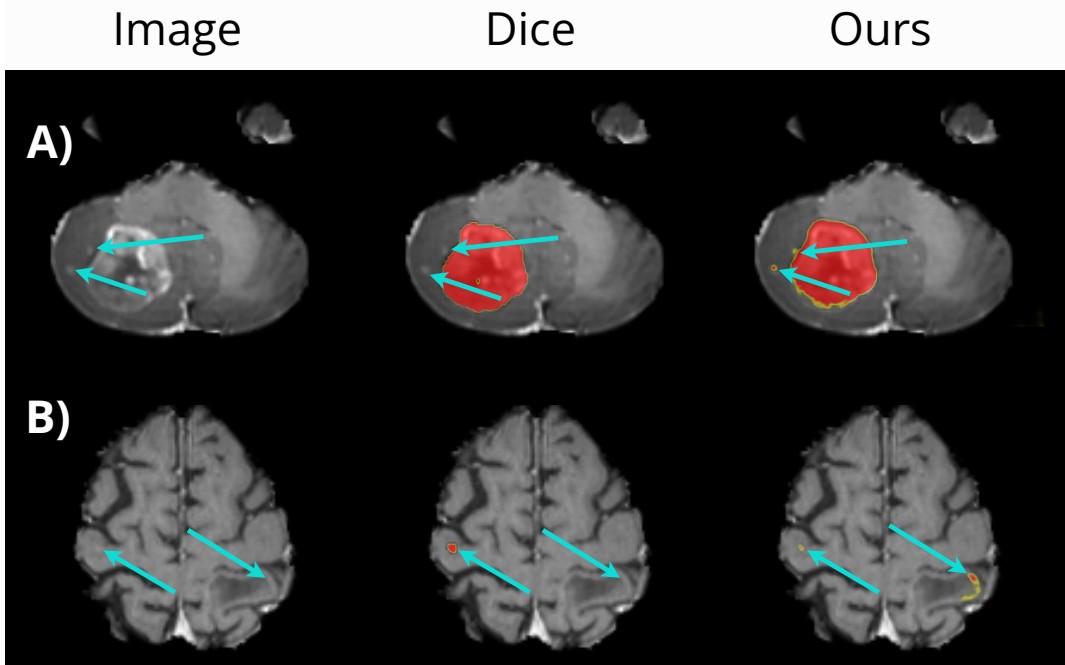

Figure 3: Visualization of the predicted probability maps as heat maps on the BraTS dataset (best viewed zoomed in). Yellow and red indicate medium and high foreground probability, respectively. Blue arrows indicate regions of overconfidence of the Dice model, while our approach exhibits well-calibrated predictions. The Dice model overconfidently predicts background (A, both arrows and B, right arrow) and foreground (B, left arrow).

On the 2D datasets, the proposed surgery yields substantial calibration gains. For INbreast, where standard losses often exhibit high instability due to the dataset's inherent challenges (foreground size variability, low foreground-to-background differences, limited sample size), our method improves training, recovering DSC performance up to 74%. We furthermore observe strong calibration performance of the CE loss, stemming from its optimal calibration properties as described in Section 3.1. However, this comes at the cost of low predictive performance, especially when facing highly imbalanced datasets, such as INbreast (DSC of 66%). On FIVES, where DSC scores are generally high (88.03%), our method maintains segmentation accuracy while drastically reducing ECE and MCE.

In the 3D domain, which presents challenges related to volumetric imbalance and label noise, our approach consistently yields better-calibrated models without compromising segmentation accuracy. While all models trained with losses containing Dice components achieve comparable DSC on BraTS and KiTS, our gradient surgery majorly reduces the MCE and NLL compared to the baseline losses. The Dice++ is notably strong on our 3D datasets; however, it is overall still inferior to the proposed gradient surgery, particularly regarding NLL and ECE. Similarly to our 2D experiments, CE-based loss functions (CE,

Table 3: Ablation study on the exponential parameter $n$ for TunableGradSym on FIVES dataset (512×512 resolution). All other hyperparameters held constant. **Bold** indicates best achieved scores and underline indicates second best results.

| Exponential $n$ | NLL ↓ | ECE ↓ | MCE ↓ | Brier ↓ | DSC (%) ↑ |
|---|---|---|---|---|---|
| 1 | 0.0565 | 0.0053 | 0.0487 | 0.0143 | 87.26 |
| 2 | 0.0541 | 0.0048 | 0.0494 | 0.0137 | 87.62 |
| 5 | 0.0535 | 0.0046 | 0.0493 | 0.0137 | 87.76 |
| 20 | **0.0528** | 0.0043 | 0.0476 | **0.0136** | **87.84** |
| 40 | **0.0528** | 0.0042 | **0.0452** | **0.0136** | **87.84** |
| 60 | 0.0529 | 0.0042 | 0.0480 | **0.0136** | 87.80 |
| 80 | 0.0531 | 0.0041 | 0.0457 | 0.0137 | 87.76 |
| 100 | 0.0538 | 0.0043 | 0.0474 | 0.0137 | 87.77 |
| 200 | 0.0533 | **0.0040** | **0.0452** | 0.0138 | 87.75 |
| 1000 | 0.0529 | 0.0043 | 0.0485 | **0.0136** | 87.72 |

SVLS, NACL) yield well-calibrated models that show weaker Dice performances because of the datasets' high region-imbalance. Combining these losses with a Dice component drastically improves DSC scores, while having an adverse effect on calibration. Notably, NACL shows poor calibration performance when evaluated on all pixels because it is underconfident in background regions. When evaluating on active regions alone (Appendix H), NACL yields calibration comparable to our method.

### 4.2. Ablation on exponential decline factor

To investigate the impact of the exponential $n$ contained in the multiplicative terms $((1 - (1 - p)^n)$ and $(1 - p^n))$ we perform an ablation study on the fives dataset with $n \in \{1, 2, 5, 20, 40, 60, 80, 100, 200, 1000\}$ and other hyperparameters fixed. The experiment shows that calibration and region overlap performance increase until $n = 20$ (see Table 3). For larger $n$, only minor differences in all metrics are observable, displaying robust performance across different values for the exponential $n$, above a certain threshold.

### 5. Conclusion

In this work, we theoretically analyze the partial derivatives of widespread region-based loss functions and show their formal connection to network calibration. We identify how the Dice/Tversky loss is incentivized to produce overconfident predictions and propose gradient surgery as a simple solution. This "surgery" combines the benefits of gradients that scale with error magnitude with robustness to region imbalance. Instead of relying on a scalar loss, we directly define vector fields at the level of the logits as loss surrogates and prove how they cannot be formalized as scalar loss functions. While this comes at the expense of desirable theoretical guarantees due to the non-conservative nature of the vector fields, we

theoretically and empirically demonstrate that our defined vector fields possess favorable properties for model training. Our method drastically improves calibration metrics across diverse medical segmentation datasets in 2D and 3D, including metastasis segmentation, where calibrated outputs provide valuable insights into borders and potential emergence of metastasis. Future work should focus on two directions: first, deriving theoretical bounds for the stability of such non-conservative gradient fields; second, exploring the utility of better-calibrated networks in clinical practice. Ultimately, this approach provides a generalizable mechanism for training uncertainty-aware segmentation networks, a prerequisite for trustworthy clinical decision support.

## Acknowledgments

This work was partially supported by the German Federal Ministry of Research, Technology, and Space (BMFTR) as part of the Software Campus 3.0 (TU München) under grant number 01IS23069.

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

## Appendix A. Proof of the non-existence of a corresponding scalar loss function

A corresponding scalar function only exists for conservative vector fields. We show that the vector field $\mathcal{F}(\mathbf{z})$ is non-conservative and therefore no corresponding scalar loss function exists. For $\mathcal{F}(\mathbf{z})$ to be conservative, there must exist a potential function $L$ (the desired loss function) such that:

$$\frac{\partial^2 L}{\partial z_i \partial z_k} = \frac{\partial^2 L}{\partial z_k \partial z_i}$$

With the desired partial derivatives, we have

$$\frac{\partial L}{\partial z_i} = \frac{p_i(1 - p_i)}{(y_i p_i + (1 - y_i)(1 - p_i))} \frac{2y_i(P + Y + \varepsilon) - (2I + \varepsilon)}{(P + Y + \varepsilon)^2},$$

and

$$\frac{\partial L}{\partial z_k} = \frac{p_k(1 - p_k)}{(y_k p_k + (1 - y_k)(1 - p_k))} \frac{2y_k(P + Y + \varepsilon) - (2I + \varepsilon)}{(P + Y + \varepsilon)^2},$$

with the second-order partial derivatives:

$$\frac{\partial^2 L}{\partial z_k \partial z_i} = \frac{p_i(1 - p_i)}{(y_i p_i + (1 - y_i)(1 - p_i))} \cdot \left[\frac{-2y_i}{(P + Y + \varepsilon)^2} - \frac{2y_k(P + Y + \varepsilon) - 2(2I + \varepsilon)}{(P + Y + \varepsilon)^3}\right] \cdot p_k(1 - p_k)$$

$$\frac{\partial^2 L}{\partial z_i \partial z_k} = \frac{p_k(1 - p_k)}{(y_k p_k + (1 - y_k)(1 - p_k))} \cdot \left[\frac{-2y_k}{(P + Y + \varepsilon)^2} - \frac{2y_i(P + Y + \varepsilon) - 2(2I + \varepsilon)}{(P + Y + \varepsilon)^3}\right] \cdot p_i(1 - p_i)$$

taking e.g. $y_i = y_k = 0$

$$\frac{\partial^2 L}{\partial z_k \partial z_i} = p_i \frac{-2(2I + \varepsilon)}{(P + Y + \varepsilon)^3} \cdot p_k(1 - p_k)$$

$$\frac{\partial^2 L}{\partial z_i \partial z_k} = p_k \frac{-2(2I + \varepsilon)}{(P + Y + \varepsilon)^3} \cdot p_i(1 - p_i)$$

$$\frac{\partial^2 L}{\partial z_k \partial z_i} = \frac{\partial^2 L}{\partial z_i \partial z_k}$$

only if

$$1 - p_k = 1 - p_i.$$

This requires $p_k = p_i$ and is obviously not true for arbitrary $p_k$ and $p_i$.

## Appendix B. Vector/gradient fields of different loss functions

Figure 4 visualizes the gradient field w.r.t to the logits $\mathbf{z}$ for different loss functions, in addition to our proposed vector field. The fields are depicted in probability space for two variables $p_1$ and $p_2$ for better visualization, although the vector represents the gradient derivatives w.r.t. to the logits. For losses influenced by global statistics (all but CE), we assume an imbalanced example with 2 foreground voxels ($y = 1$) and 98 background voxels ($y = 0$). With assumed probability values of 0.8 for foreground voxels and 0.1 for background voxels. The displayed gradient/vector fields add 2 additional voxels $y_1 = 1$ and $y_2 = 2$, and show the gradient on their logits for different $p_1$ and $p_2$ values, while the probabilities/logits for the other 100 voxels stay unchanged.

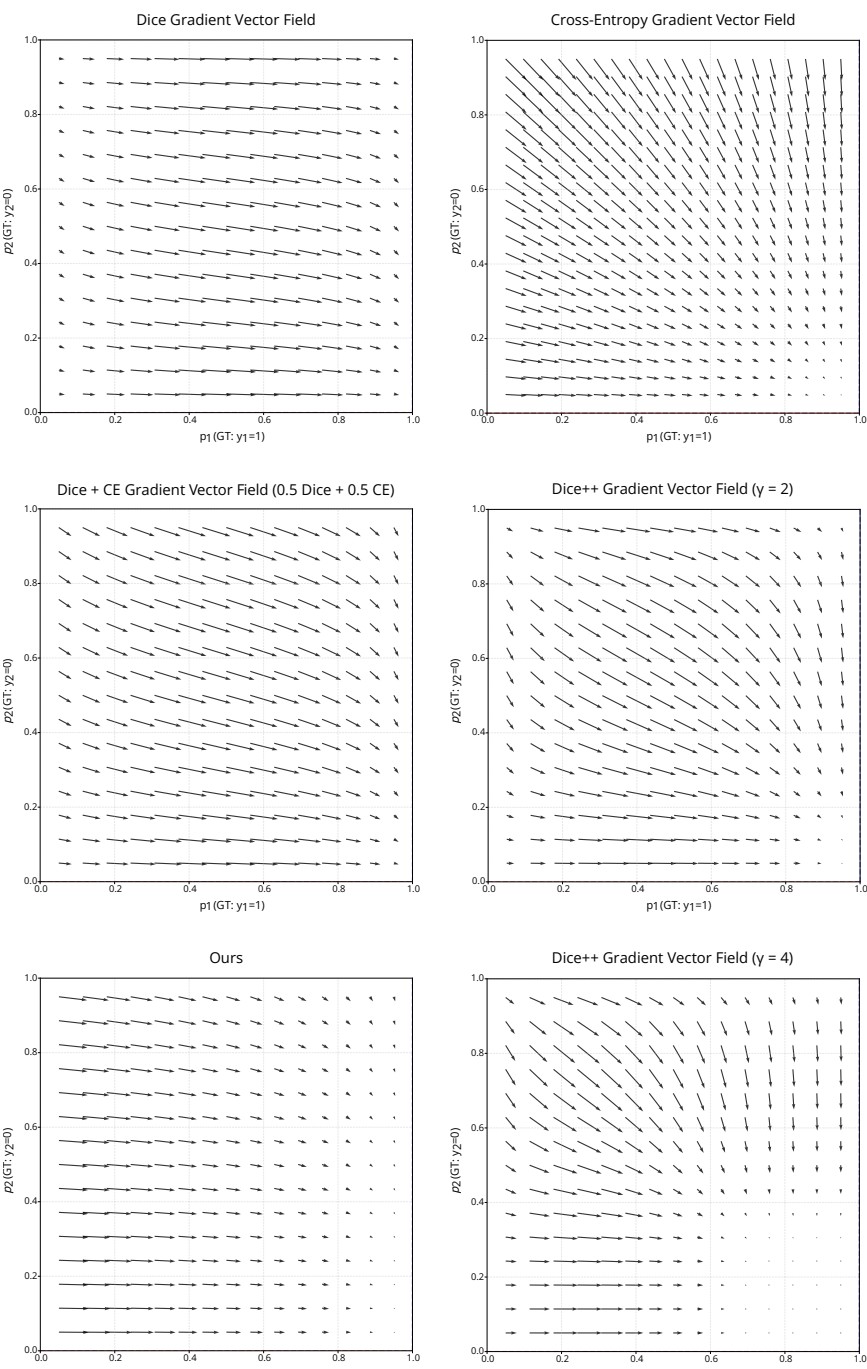

Figure 4: Gradient/vector fields w.r.t. logits of different loss functions and our proposed vector field. For better visualization, the axes are displayed as probabilities $p$ instead of logits $z$.

## Appendix C. Experimentation details

### DATASETS

The INbreast dataset (Moreira et al., 2012) contains 107 images with Masses. We separate 22 ($\sim 20\%$ of total) images for the test set. Validation metrics for model selection are calculated on 17 ($= 20\%$ of remaining) of the remaining images. We resize the images to a resolution of $512x \times 512$ for model training.

The FIVES dataset (Jin et al., 2022) contains 800 fundus images with vessel annotations. We rescale images to a resolution of $1024 \times 1024$, and train on the center and evaluate on the central patch of $512 \times 512$ voxels. We separate 200 images ($= 20\%$ of total) for testing. Of the remaining data, we use 120 ($= 20\%$ of remaining) images as validation set.

The original BraTS Metastasis dataset (Maleki et al., 2025) comprises a retrospective collection of 1296 pre- and post-treatment brain metastases, labeled in four classes: nonenhancing tumor core, FLAIR hyperintensity, enhancing tumor, and resection cavity. Each sample has four input channels (T1, T1c, T2, FLAIR). We use a random, representative subset of 156 cases for training, 44 for validation, and 251 for testing. As input, we use only the T1c scan, disregarding the others. The images vary in size, orientation, and spacing. We preprocess each image in a nnUNet-style fashion (Isensee et al., 2021) with reorientation to RAS+, resampling to an isotropic spacing of $1mm$, and z-score normalization. The resulting volumes have a median shape of $[141 \times 175 \times 142]$. We convert the labels to a binary format, where enhancing and non-enhancing tumor tissue is foreground and FLAIR hyperintensity, resection cavity, and healthy tissue is background. The lesions account for 0.17% of the total voxels with a standard deviation of 0.29% per sample. During training, we extract a random patch of size $[80 \times 96 \times 80]$ with a foreground oversample ratio of 0.33.

The KiTS dataset (Heller et al., 2019) comprises 489 abdominal CT scans where kidneys, renal tumors, and renal cysts are labeled. The images vary in size, orientation, and spacing. We preprocess each image in a nnUNet-style fashion (Isensee et al., 2021) with reorientation to RAS+, resampling to an isotropic spacing of $1.5mm$, and intensity-clipping to the $[0.5, 99.5]$ percentiles (i.e., $[-58, 302]$ HU) followed by z-score normalization. We extract ROIs of varying sizes with a median of $[218 \times 130 \times 160]$ around both kidneys. We convert the labels to a binary format, where the tumors are foreground and the kidneys, cysts, and the rest are background. The tumors account for 0.68% of the total voxels with a standard deviation of 1.18% per sample. We stratify the complete dataset into 192 training, 50 validation, and 247 test sets, which are balanced in terms of size and number of tumors. During training, we extract 8 random patches per sample, with a foreground oversample ratio of 0.45 and a fixed size of $[96 \times 96 \times 80]$ for each patch.

### LOSS HYPERPARAMETERS

For the Tversky loss, we set the $\alpha$ parameter ($\beta = 1 - \alpha$) as a hyperparameter. For Combo loss (Taghanaki et al., 2019), we fix the weighting to 0.5 (Isensee et al., 2021). For Dice++, we set the $\gamma$ parameter to 2 (Yeung et al., 2023). For SVLS, we set the kernel size to 3 and use $\sigma$ as a hyperparameter with possible settings of 1, 2, and 3 (Islam and Glocker, 2021). For NACL, we use the penalty formulation, set the balancing parameter to 0.1 ($\lambda$), the kernel size to 3, use a mean prior ($\tau$), and use an L1 penalty(Murugesan et al., 2025).

Finally, for the mL1-ACE loss, we use equal weighting with Dice loss and use 20 bins to discretize the probability space (Barfoot et al., 2024).

## Model and Training Procedure

Our 3D experiments utilize a full-resolution 3D UNet with residual units, following a pipeline heavily influenced by nnUNet (Isensee et al., 2021), particularly in terms of network size, learning rate schedule, iterations, augmentations, and optimizer. We use SGD with Nesterov momentum as an optimizer. In our hyperparameter optimization, we further optimize for weight decay, initial learning rate, and momentum. At test time, we do sliding window inference on the complete volume with an overlap ratio of 0.5 and Gaussian weighting.

## Training curves stability

Figure 5 shows training and validation curves with different optimizers.

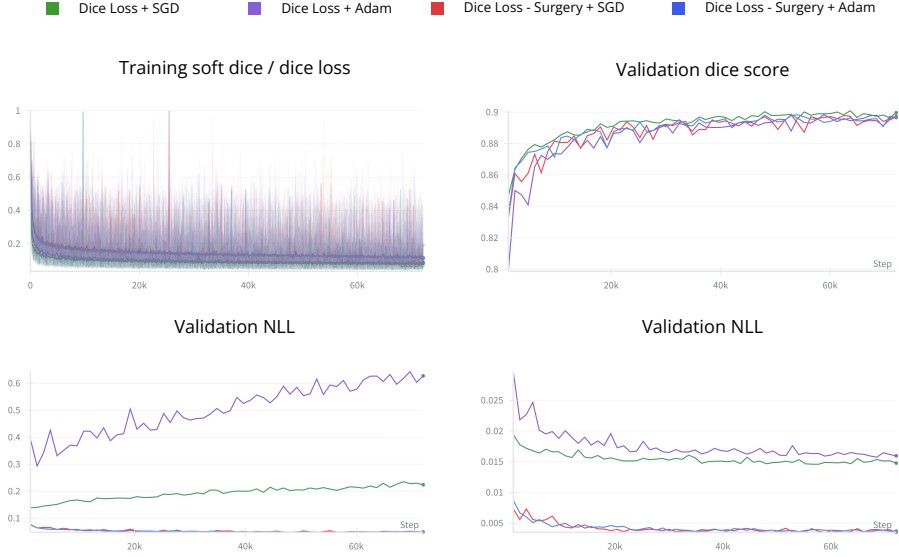

Figure 5: Training curves with SGD+Nesterov Momentum and Adam optimizers for a dice loss training and a model with the same hyperparameters trained with our dice surgery training.objective

## Appendix D. Implementation details for custom vector fields

In practice, we implement our custom ("gradient") vector field for the different region-based losses by overwriting the backward pass of the softmax activation. The forward pass through the softmax remains unchanged. We keep the partial derivatives of the Loss w.r.t. the probabilities and exchange the partial derivative of the probabilities w.r.t. the logits to reflect our desired vector field, e.g., by replacing $p(1-p)$ with $|y-p|$ and adding the sharp decline term close to $p = 0$ and $p = 1$, $(1-(1-p)^n)$ and $(1-p^n)$. The actual implementation is shown in the following listing.

```python
class GradSurgeSoftmax(torch.autograd.Function):
    @staticmethod
    @custom_fwd(device_type="cuda", cast_inputs=torch.float32)
    def forward(ctx, logits, targets, exponential_correction=None):
        probs = torch.softmax(logits, dim=1)

        error = torch.abs(probs - targets)

        if exponential_correction is not None:
            # Applying the correction term
            error_weight = 0.25 * error * (1 - torch.pow(error,
    exponential_correction)) * (1 - torch.pow(1 - error,
    exponential_correction))
        else:
            error_weight = 0.25 * error

        ctx.save_for_backward(error_weight)
        return probs

    @staticmethod
    @custom_bwd(device_type="cuda")
    def backward(ctx, grad_output):
        error_weight, = ctx.saved_tensors

        grad_output = grad_output.to(error_weight.dtype)

        # Assuming binary segmentation (background vs foreground)
        weight = error_weight[:, 1:2]
        grad_p_bg = grad_output[:, 0:1]
        grad_p_fg = grad_output[:, 1:2]

        coupling = (grad_p_fg - grad_p_bg)

        grad_logits_bg = -weight * coupling
        grad_logits_fg = weight * coupling

        return torch.cat([grad_logits_bg, grad_logits_fg], dim=1), None,
    None, None
```

Listing 1: Implementation of GradSurgeSoftmax

## Appendix E. Dice++ gradient

The Dice ++ loss

$$DSC++ = 1 - \frac{2\sum_{i=1}^{N} p_i y_i + \epsilon}{2\sum_{i=1}^{N} p_i y_i + \sum_{i=1}^{N}(p_i(1-y_i))^\gamma + \sum_{i=1}^{N}((1-p_i)y_i)^\gamma + \epsilon}$$

was proposed to resolve the calibration issues of the dice loss by introducing a focus $\gamma$ on false positives and false negatives.

$$\frac{\partial L_{DSC++}}{\partial p_i^{fg}} = \frac{-2[\gamma(1-p_i)^{\gamma-1}I + FP^\gamma + FN^\gamma]}{(2I + 2p_i + (1-p_i)^\gamma + FP_{-i}^\gamma + FN_{-i}^\gamma)^2}$$

$$\frac{\partial L_{DSC++}}{\partial p_i^{bg}} = \frac{2\gamma p_i^{\gamma-1} 2I}{(2I + p_i^\gamma + FP_{-i}^\gamma + FN_{-i}^\gamma)^2}$$

Exactly, for $\gamma = 2$, the partial derivative of the Dice++ loss w.r.t. the probabilities depends linearly on the error as for MSE loss on the probabilities ($\partial L_{DSC++}/\partial p_i^{fb} = 2p_i$) while the global term for $y = 0$ and $y = 1$ is the most similar to the original dice loss at $\gamma = 2$ compared to higher $\gamma$'s. Which we identify as the reason for the optimal performance in terms of Dice and calibration metrics for $\gamma = 2$.

Besides this desired property, the $\gamma$ parameter introduces a vast downscaling of the gradient for samples with large proportions of false positives and false negatives. This can be problematic for cases where (1) the foreground regions have drastically different sizes and, in connection to that, drastically different values for false positives and false negatives, and (2) for cases where learning from "hard" examples characterized through high false positive and false negative rates is crucial. On the contrary, previous works also showed that in some cases, focus on easy examples can be beneficial for segmentation performance (Abraham and Khan, 2019).

## Appendix F. Partial derivative function visualization

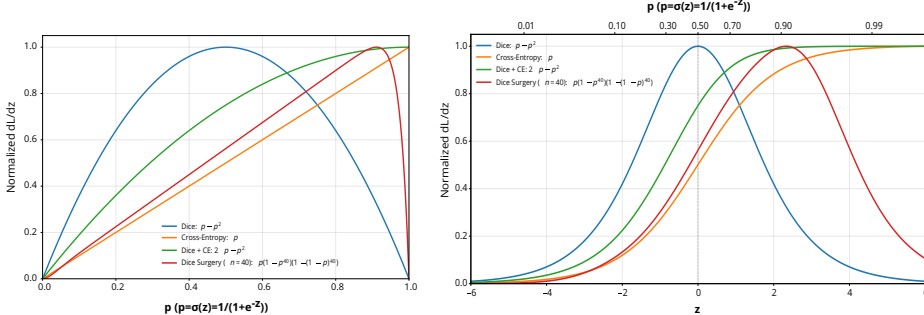

Figure 6: Visualization of the normalized partial derivatives derived from different loss functions.

## Appendix G. Qualitative examples

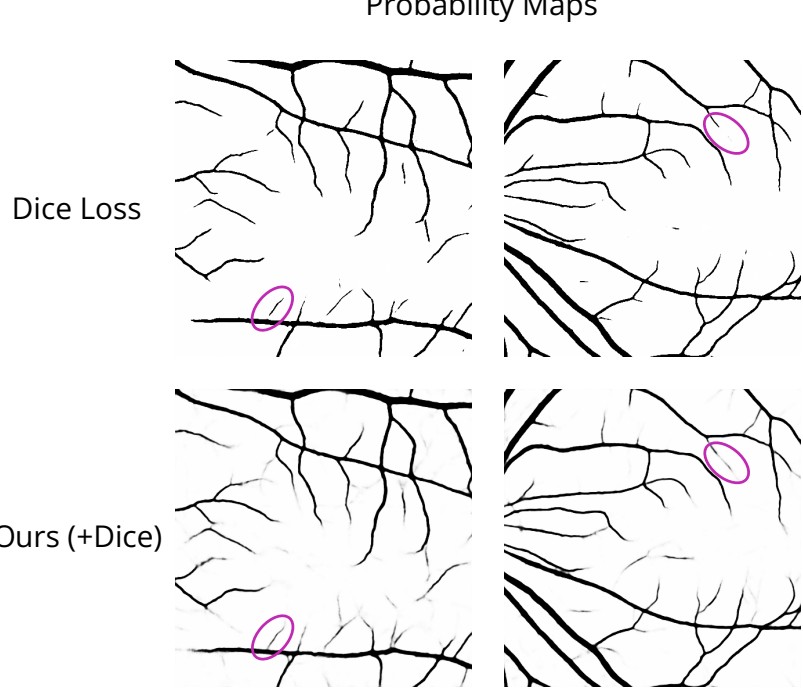

Figure 7: Comparison of the probability maps for two samples from a Dice loss trained model and our adapted Dice vector field approach. The overconfidence phenomenon on the Dice probabilities is apparent, with probabilities almost appearing binarized. In comparison, our method has probabilities other than 0 and 1, which often reflect plausible vascular courses (e.g., purple ellipses) that are missing in the probability maps of the Dice loss-trained model.

## Appendix H. Active region evaluation

Tables 4 and 5 show an evaluation of the calibration metrics on the "active" foreground region defined as the union of target and (binarized) prediction foreground regions (Murugesan et al., 2025). Evaluation only on the active region results in a stark difference for the absolute values on calibration metrics. However, we see that the trends in the improvement achieved through our method remain unchanged.

Table 4: Average test set performance of the 5 best runs out of 25 (selected through validation dice scores) on the INbreast and FIVES datasets with active region calibration result. The active region is defined as the union of the target foreground and the binarized prediction foreground. Best results are displayed in **bold**, second-best results are underlined. Significantly better performance is highlighted through a $^*$ using the 0.01 significance level.

| | Method | NLL ↓ | ECE ↓ | MCE ↓ | Brier ↓ | Dice (%) ↑ |
|---|---|---|---|---|---|---|
| INbreast (Active) | Cross Entropy | $1.1372$ $_{\pm0.1571}$ | $0.3229$ $_{\pm0.0502}$ | $0.5940$ $_{\pm0.2130}$ | $0.3085$ $_{\pm0.0290}$ | $66.20$ $_{\pm3.57}$ |
| | NACL | $1.2444$ $_{\pm0.2337}$ | $0.3167$ $_{\pm0.0427}$ | $0.6731$ $_{\pm0.2730}$ | $0.3064$ $_{\pm0.0347}$ | $68.22$ $_{\pm3.57}$ |
| | NACL + Dice | $2.0573$ $_{\pm0.5311}$ | $0.3367$ $_{\pm0.0332}$ | $0.5938$ $_{\pm0.0380}$ | $0.3321$ $_{\pm0.0297}$ | $72.07$ $_{\pm2.57}$ |
| | SVLS | $1.1109$ $_{\pm0.1555}$ | $0.3089$ $_{\pm0.0295}$ | $0.6844$ $_{\pm0.2495}$ | $0.2908$ $_{\pm0.0233}$ | $68.35$ $_{\pm2.90}$ |
| | SVLS + Dice | $1.7828$ $_{\pm0.3316}$ | $0.3535$ $_{\pm0.0367}$ | $0.5863$ $_{\pm0.0428}$ | $0.3470$ $_{\pm0.0304}$ | $68.76$ $_{\pm2.70}$ |
| | Dice++ | $1.2198$ $_{\pm0.0628}$ | $0.3207$ $_{\pm0.0251}$ | $0.5231$ $_{\pm0.0293}$ | $0.3137$ $_{\pm0.0166}$ | $67.51$ $_{\pm3.17}$ |
| | CE + Dice | $2.1938$ $_{\pm0.3776}$ | $0.3411$ $_{\pm0.0270}$ | $0.5930$ $_{\pm0.0353}$ | $0.3355$ $_{\pm0.0256}$ | $71.76$ $_{\pm2.10}$ |
| | CE + Dice - Surgery | $\mathbf{0.8807}^*$ $_{\pm0.0926}$ | $\underline{0.2427}^*$ $_{\pm0.0322}$ | $\underline{0.4283}^*$ $_{\pm0.0388}$ | $\mathbf{0.2470}^*$ $_{\pm0.0250}$ | $\mathbf{74.39}$ $_{\pm3.34}$ |
| | Dice | $3.2670$ $_{\pm1.3015}$ | $0.3996$ $_{\pm0.0433}$ | $0.6189$ $_{\pm0.0613}$ | $0.3940$ $_{\pm0.0401}$ | $65.06$ $_{\pm3.31}$ |
| | Dice - Surgery | $\underline{0.9670}$ $_{\pm0.1173}$ | $\mathbf{0.2397}^*$ $_{\pm0.0313}$ | $\mathbf{0.4228}^*$ $_{\pm0.0548}$ | $\underline{0.2496}^*$ $_{\pm0.0257}$ | $\underline{74.34}^*$ $_{\pm2.77}$ |
| | Tversky | $3.6811$ $_{\pm0.6354}$ | $0.3981$ $_{\pm0.0238}$ | $0.6365$ $_{\pm0.0217}$ | $0.3912$ $_{\pm0.0213}$ | $67.63$ $_{\pm2.89}$ |
| | Tversky - Surgery | $1.0767^*$ $_{\pm0.1626}$ | $0.2842^*$ $_{\pm0.0405}$ | $0.4754^*$ $_{\pm0.0527}$ | $0.2808^*$ $_{\pm0.0307}$ | $69.34$ $_{\pm3.77}$ |
| FIVES (Active) | Cross Entropy | $0.5448$ $_{\pm0.0121}$ | $0.1265$ $_{\pm0.0017}$ | $0.2856$ $_{\pm0.0088}$ | $0.1465$ $_{\pm0.0017}$ | $87.54$ $_{\pm0.22}$ |
| | NACL | $0.5324$ $_{\pm0.0071}$ | $\underline{0.1253}$ $_{\pm0.0021}$ | $\underline{0.2844}$ $_{\pm0.0109}$ | $0.1450$ $_{\pm0.0012}$ | $87.60$ $_{\pm0.14}$ |
| | NACL + Dice | $0.7641$ $_{\pm0.0341}$ | $0.1597$ $_{\pm0.0033}$ | $0.4003$ $_{\pm0.0106}$ | $0.1634$ $_{\pm0.0028}$ | $87.94$ $_{\pm0.11}$ |
| | Dice++ | $0.5365$ $_{\pm0.0039}$ | $0.1343$ $_{\pm0.0014}$ | $0.3286$ $_{\pm0.0127}$ | $0.1452$ $_{\pm0.0010}$ | $87.97$ $_{\pm0.07}$ |
| | ACE | $0.8581$ $_{\pm0.0274}$ | $0.1514$ $_{\pm0.0044}$ | $0.2927$ $_{\pm0.0116}$ | $0.1688$ $_{\pm0.0052}$ | $85.96$ $_{\pm0.45}$ |
| | CE + Dice | $0.7682$ $_{\pm0.0160}$ | $0.1594$ $_{\pm0.0025}$ | $0.3972$ $_{\pm0.0106}$ | $0.1634$ $_{\pm0.0020}$ | $\underline{87.99}$ $_{\pm0.16}$ |
| | CE + Dice - Surgery | $0.5340^*$ $_{\pm0.0264}$ | $\mathbf{0.1249}^*$ $_{\pm0.0050}$ | $\mathbf{0.2815}^*$ $_{\pm0.0110}$ | $\underline{0.1447}^*$ $_{\pm0.0027}$ | $87.70$ $_{\pm0.09}$ |
| | Dice | $2.7751$ $_{\pm0.3224}$ | $0.1947$ $_{\pm0.0050}$ | $0.5847$ $_{\pm0.0264}$ | $0.1940$ $_{\pm0.0049}$ | $\mathbf{88.03}$ $_{\pm0.25}$ |
| | Dice - Surgery | $\underline{0.5271}^*$ $_{\pm0.0247}$ | $0.1283^*$ $_{\pm0.0050}$ | $0.2999^*$ $_{\pm0.0131}$ | $0.1451^*$ $_{\pm0.0024}$ | $87.60$ $_{\pm0.12}$ |
| | Tversky | $2.5810$ $_{\pm0.2671}$ | $0.1969$ $_{\pm0.0018}$ | $0.5727$ $_{\pm0.0081}$ | $0.1960$ $_{\pm0.0019}$ | $87.77$ $_{\pm0.21}$ |
| | Tversky - Surgery | $\mathbf{0.5189}^*$ $_{\pm0.0155}$ | $0.1258^*$ $_{\pm0.0029}$ | $0.2953^*$ $_{\pm0.0150}$ | $\mathbf{0.1432}^*$ $_{\pm0.0021}$ | $87.79$ $_{\pm0.25}$ |

Table 5: Test-set performance on the KiTS dataset (Heller et al., 2019) with active region calibration results. The active region is defined as the union of the target foreground and the binarized prediction foreground. Best results are displayed in **bold**, second-best results are underlined. Better performance for standard losses vs. altered losses is highlighted through *italics*.

|  | Method | NLL $\downarrow$ | ECE $\downarrow$ | MCE $\downarrow$ | Brier $\downarrow$ | DSC (%) $\uparrow$ |
|---|---|---|---|---|---|---|
| | CE | 1.6627 | 0.3306 | 0.5117 | 0.6870 | 64.79 |
| | SVLS | **0.9213** | 0.2615 | 0.4867 | 0.5249 | 70.68 |
| | SVLS + Dice | 2.5133 | 0.2994 | 0.5438 | 0.5969 | 75.82 |
| | NACL | 0.9398 | 0.3033 | 0.4730 | 0.6263 | 64.48 |
| KiTS Tumor | NACL + Dice | 1.3766 | 0.2981 | 0.5567 | 0.5918 | 74.56 |
| | Dice++ | 6.3755 | 0.2580 | 0.4445 | 0.5244 | 75.57 |
| | CE + Dice | 8.9081 | 0.3042 | 0.5664 | 0.6041 | 75.77 |
| | CE + Dice - Surgery | *1.2973* | **0.2344** | **0.4132** | **0.4830** | **76.75** |
| | Dice | 5.5300 | 0.3148 | 0.6309 | 0.6253 | *76.62* |
| | Dice - Surgery | *1.2977* | *0.2611* | *0.4393* | *0.5193* | 74.01 |
| | Tversky | 7.3410 | 0.3506 | 0.6110 | 0.6961 | 73.10 |
| | Tversky - Surgery | *1.5339* | *0.2671* | *0.4723* | *0.5258* | *73.87* |

## Appendix I. Effect on different foreground sizes

Table 6 summarizes the detection performance and ECE for different tumor sizes on the Kits dataset. A tumor is considered detected if at least one voxel within its label area is correctly predicted as foreground. Very small foreground components (below $4cm^3$) are not included in the comparison as they are assumed to constitute label noise (Berger et al., 2025). We observe consistent calibration improvements when applying our proposed gradient field surgery across all tumor sizes. Furthermore, our approach improves the detection of small tumors slightly more than it does for large and very large tumors. We hypothesize that improved calibration is most beneficial for small tumors, where model uncertainty is naturally higher; in these borderline cases, accurate probability estimates are critical for successful detection because they help push the predicted probabilities of these difficult cases close to the detection threshold, thereby improving detection.

ECE values are calculated on the lesion foreground pixels and averaged across lesions of each size category.

Table 6: Detection rate (Det) and Expected Calibration Error (ECE) across different lesion sizes and loss functions.

| Lesion Size | Metric | Dice++ | CE + Dice | CE + Dice Surgery | Dice | Dice Surgery | Tversky | Tversky Surgery |
|---|---|---|---|---|---|---|---|---|
| Small | Det. | 0.9221 | 0.8961 | 0.9481 | 0.9091 | 0.9091 | 0.8961 | 0.9481 |
|  | ECE | 0.2272 | 0.3112 | 0.2172 | 0.2764 | 0.2182 | 0.3586 | 0.1664 |
| Large | Det. | 0.9872 | 0.9615 | 0.9359 | 0.9615 | 0.9744 | 0.9615 | 0.9487 |
|  | ECE | 0.0976 | 0.1607 | 0.1279 | 0.1611 | 0.0991 | 0.2128 | 0.1170 |
| Very Large | Det. | 1.000 | 0.9744 | 0.9871 | 0.9872 | 0.9872 | 1.000 | 0.9744 |
|  | ECE | 0.055 | 0.0949 | 0.0610 | 0.0949 | 0.0589 | 0.1134 | 0.0558 |

## Appendix J. Experiments with transformer architecture

In addition to the main experiments (nn-unet style training), we perform experiments with the state-of-the-art Primus transformer architecture (Wald et al., 2025) for image segmentation. The results in 7 show that vector field surgery also works in combination with transformer architectures, yielding notably better calibration scores. However, the overall performance of the transformer approach was poor. In medical image segmentation, convolutional approaches have proven more effective in numerous extensive evaluation studies (Isensee et al., 2021, 2024; Wald et al., 2025).

Table 7: Results on the KiTS tumor segmentation dataset, using the PRIMUS (Wald et al., 2025) transformer architecture.

| Method | NLL ↓ | ECE ↓ | MCE ↓ | Brier ↓ | DSC (%) ↑ |
|---|---|---|---|---|---|
| Dice + CE | 0.0188 | 0.0031 | 0.2308 | 0.0078 | **67.62** |
| Dice + CE - Surgery | **0.0156** | **0.0021** | **0.1173** | **0.0071** | 66.86 |

## Appendix K. Effect on logit values

Table 8 shows the average logit values in target foreground and background regions. Our method reduces the logit distance and the absolute value of the logits. Especially for the FIVES dataset, standard region-based losses show very large logit values, indicating overconfidence. Reduced logit magnitudes were found to result in improved calibration scores in earlier works on label smoothing and logit constraints (Müller et al., 2019; Murugesan et al., 2025). The analysis provides direct evidence that the vector field intervention is effective in resolving the overconfidence problem of region-based losses.

Table 8: Logit magnitude analysis on INbreast and FIVES datasets (average of top 5 runs ± standard deviation).

| | Method | Target FG | | Target BG | | Dice (%) ↑ |
|---|---|---|---|---|---|---|
| | | FG Logit | BG Logit | FG Logit | BG Logit | |
| INbreast | CE + Dice | $3.51$ $_{\pm 2.23}$ | $-1.07$ $_{\pm 1.27}$ | $-5.39$ $_{\pm 0.76}$ | $6.31$ $_{\pm 0.71}$ | $71.76$ $_{\pm 2.10}$ |
| | CE + Dice - Surgery | $1.52$ $_{\pm 0.90}$ | $-1.33$ $_{\pm 0.89}$ | $-4.31$ $_{\pm 0.53}$ | $4.93$ $_{\pm 0.47}$ | $74.39$ $_{\pm 3.34}$ |
| | Dice | $5.18$ $_{\pm 2.34}$ | $-1.42$ $_{\pm 3.43}$ | $-5.26$ $_{\pm 1.08}$ | $6.18$ $_{\pm 0.81}$ | $65.06$ $_{\pm 3.31}$ |
| | Dice - Surgery | $1.87$ $_{\pm 1.89}$ | $-0.67$ $_{\pm 1.48}$ | $-4.41$ $_{\pm 0.40}$ | $5.53$ $_{\pm 0.65}$ | $74.34$ $_{\pm 2.77}$ |
| | Tversky | $5.97$ $_{\pm 0.97}$ | $-1.79$ $_{\pm 2.55}$ | $-5.39$ $_{\pm 0.69}$ | $6.34$ $_{\pm 0.38}$ | $67.63$ $_{\pm 2.89}$ |
| | Tversky - Surgery | $2.28$ $_{\pm 1.67}$ | $-0.33$ $_{\pm 1.76}$ | $-4.52$ $_{\pm 0.81}$ | $5.58$ $_{\pm 0.54}$ | $69.34$ $_{\pm 3.77}$ |
| FIVES | CE + Dice | $3.71$ $_{\pm 1.13}$ | $-4.18$ $_{\pm 1.07}$ | $-4.08$ $_{\pm 0.06}$ | $4.57$ $_{\pm 0.14}$ | $87.99$ $_{\pm 0.16}$ |
| | CE + Dice - Surgery | $2.71$ $_{\pm 1.03}$ | $-2.51$ $_{\pm 1.02}$ | $-3.19$ $_{\pm 0.24}$ | $3.71$ $_{\pm 0.22}$ | $87.70$ $_{\pm 0.09}$ |
| | Dice | $26.42$ $_{\pm 13.03}$ | $-29.12$ $_{\pm 18.00}$ | $-7.80$ $_{\pm 1.18}$ | $7.74$ $_{\pm 0.59}$ | $88.03$ $_{\pm 0.25}$ |
| | Dice - Surgery | $3.66$ $_{\pm 0.18}$ | $-1.91$ $_{\pm 0.45}$ | $-2.92$ $_{\pm 0.28}$ | $3.52$ $_{\pm 0.19}$ | $87.60$ $_{\pm 0.12}$ |
| | Tversky | $18.79$ $_{\pm 11.26}$ | $-18.51$ $_{\pm 12.57}$ | $-6.82$ $_{\pm 1.12}$ | $7.03$ $_{\pm 0.79}$ | $87.77$ $_{\pm 0.21}$ |
| | Tversky - Surgery | $3.40$ $_{\pm 0.99}$ | $-2.36$ $_{\pm 0.94}$ | $-2.96$ $_{\pm 0.12}$ | $3.49$ $_{\pm 0.16}$ | $87.79$ $_{\pm 0.25}$ |

