# OpenReview forum: "Beyond scalar losses: calibrating segmentation models via gradient vector field surgery"
_MIDL.io/2026/Conference — MIDL 2026 Poster_

### Official Review · Reviewer_JCqw · 2026-01-10

**Confidence:** 3
**Preliminary Rating:** 3
**Final Rating:** 4

**Summary:**

The paper proposes a gradient-based loss component to improve the segmentation performance for medical images. It uses a partial derivative, which aligns with the prediction error magnitude. The framework is evaluated for both 2D and 3D segmentation tasks. The reasoning is backed by mathematical deductions, which are sound. The visualization of the predicted segments also looks good. However, the abstract lacks quantitative results.

**Strengths:**

1. The proposed loss component has novelty.
2. The new loss can be combined with existing losses like dice and CE to improve the segmentation performance.
3. Evaluated on both 3D and 2D datasets. Utilized benchmark datasets.

**Weaknesses:**

1. It tested the loss function with a single model Unet, which can limit generalizability.
2. It is not clear if this can be useful for transformer-based segmentation models.
3. Also, more experiments or inspections are required to understand for which cases (small lesions, large lesions, image modalities, etc) this loss function works better.

**Detailed Comments:**

The proposed loss function is to improve the segmentation, so it would be better if this loss function could be integrated with any segmentation-based model.

However, there are no experiments with different models, like transformer-based segmentation models. It would be good to see this new loss work as model, domain, and task agnostic.

**Justification Of Final Rating:**

Thank you for addressing my comments. The author added new experiment results showing the effectiveness of adding their module to transformer-based architectures. Moreover, they evaluated the utility of the proposed framework across different imaging modalities: X-ray (INBreast), MR (BraTS Metastasis), CT (KiTS Tumor), and color fundus photography (FIVES).
I am increasing my rating by 1.

**Justification Of The Preliminary Rating:**

This paper proposes a new loss component which improves the segmentation performance. The reasoning is backed by mathematical deductions, which are sound. The visualization of the predicted segments also looks good.

**Questions To Address In The Rebuttal:**

Please check the limitation and detailed comments.

---

> ### Author Response · Authors · 2026-01-25
>
> We thank you for your interesting questions and provide answers below.
>
> **R JCqw:** Can your solution be applied to other segmentation models, e.g., Transformers?
>
> **TLDR Answer:** Yes. We conducted additional experiments, described them in Appendix J of the paper, and summarized our findings below.
>
> To further investigate the utility of our gradient vector field surgery, we perform experiments using a state-of-the-art transformer architecture for medical image segmentation [1]. We added the experiment results to Appendix J of the paper and also show them below. The results indicate that vector field surgery also works in combination with transformer architectures, achieving notably better calibration scores. However, the overall performance of the transformer approach lagged behind the convolutional approach. In medical image segmentation, convolutional approaches have proven more effective in numerous extensive evaluation studies [1,2,3].
>
> | | Method | NLL ↓ | ECE ↓ | MCE ↓ | Brier ↓ | DSC (%) ↑ |
> | :--- | :--- | :---: | :---: | :---: | :---: | :---: |
> | | Dice + CE | 0.0188 | 0.0031 | 0.2308 | 0.0078 | **67.62** |
> | | Dice + CE - Surgery | **0.0156** | **0.0021** | **0.1173** | **0.0071** | 66.86 |
>
> **R JCqw:** For which image modalities and which foreground (e.g., lesion) sizes/types does our approach work best?
>
> **TLDR Answer:** Our method works across modalities and lesion types. We ran additional evaluations to test if our method behaves differently for specific foreground (i.e., tumor) sizes and found that small foregrounds benefit slightly more.
>
> Given that our method includes region-loss-specific adaptive handling of region/class-size imbalance, it performs best in cases where region-based losses are effective. This is the case in scenarios of (1) extreme class imbalance [4,5,6], e.g., in the KiTs Tumor and BraTs Metastasis segmentation tasks, and in scenarios of (2) strong region size variation, e.g., in the INBreast dataset, where mass sizes vary drastically. In cases of relatively balanced class labels combined with stable class ratios, we do not expect our method to outperform the standard cross-entropy loss. This is exactly what we observe in the FIVES dataset, where foreground is not extremely underrepresented, and foreground-background ratios are relatively constant across images.
>
> We evaluate the utility of our methods across vastly different settings, including four imaging modalities: X-ray (INBreast), MR (BraTS Metastasis), CT (KiTS Tumor), and color fundus photography (FIVES). Each dataset contains a unique task with drastically different characteristics, namely, mass segmentation in the breast, metastasis segmentation in the brain, tumor segmentation in the kidney, and vessel segmentation in the eye.
>
> | Lesion Size | Metric | Dice++ | CE + Dice | CE + Dice Surg. | Dice | Dice Surg. | Tversky | Tversky Surg. |
> | :--- | :--- | :--- | :--- | :--- | :--- | :--- | :--- | :--- |
> | **Small** | Det. | 0.9221 | 0.8961 | 0.9481 | 0.9091 | 0.9091 | 0.8961 | 0.9481 |
> | | ECE | 0.2272 | 0.3112 | 0.2172 | 0.2764 | 0.2182 | 0.3586 | 0.1664 |
> | **Large** | Det. | 0.9872 | 0.9615 | 0.9359 | 0.9615 | 0.9744 | 0.9615 | 0.9487 |
> | | ECE | 0.0976 | 0.1607 | 0.1279 | 0.1611 | 0.0991 | 0.2128 | 0.1170 |
> | **Very Large** | Det. | 1.000 | 0.9744 | 0.9871 | 0.9872 | 0.9872 | 1.000 | 0.9744 |
> | | ECE | 0.055 | 0.0949 | 0.0610 | 0.0949 | 0.0589 | 0.1134 | 0.0558 |
>
> **R JCqw:** The abstract does not mention quantitative results.
>
> We updated the abstract to briefly reference our quantitative improvements.

---

> ### Author Response · Authors · 2026-01-25
>
> **References**
>
> [1] Wald, T., Roy, S., Isensee, F., Ulrich, C., Ziegler, S., Trofimova, D., ... & Maier-Hein, K. (1835). Primus: enforcing attention usage for 3d medical image segmentation (2025). arXiv preprint arXiv:2503.01835.
>
> [2] Isensee, F., Jaeger, P. F., Kohl, S. A., Petersen, J., & Maier-Hein, K. H. (2021). nnU-Net: a self-configuring method for deep learning-based biomedical image segmentation. Nature methods, 18(2), 203-211.
>
> [3] Isensee, F., Wald, T., Ulrich, C., Baumgartner, M., Roy, S., Maier-Hein, K., & Jaeger, P. F. (2024, October). nnu-net revisited: A call for rigorous validation in 3d medical image segmentation. In International Conference on Medical Image Computing and Computer-Assisted Intervention (pp. 488-498). Cham: Springer Nature Switzerland.
>
> [4] Eelbode, T., Bertels, J., Berman, M., Vandermeulen, D., Maes, F., Bisschops, R., & Blaschko, M. B. (2020). Optimization for medical image segmentation: theory and practice when evaluating with dice score or jaccard index. IEEE transactions on medical imaging, 39(11), 3679-3690.
>
> [5] Ma, J., Chen, J., Ng, M., Huang, R., Li, Y., Li, C., ... & Martel, A. L. (2021). Loss odyssey in medical image segmentation. Medical image analysis, 71, 102035.
>
> [6] Sudre, C. H., Li, W., Vercauteren, T., Ourselin, S., & Jorge Cardoso, M. (2017, September). Generalised dice overlap as a deep learning loss function for highly unbalanced segmentations. In International Workshop on Deep Learning in Medical Image Analysis (pp. 240-248). Cham: Springer International Publishing.

---

### Official Review · Reviewer_wV9J · 2026-01-10

**Confidence:** 5
**Preliminary Rating:** 4
**Final Rating:** 5

**Summary:**

This paper addresses the critical issue of miscalibration in medical image segmentation models trained with region-based losses like Dice or Tversky loss. These losses, while robust to class imbalance, typically lead to overconfidence, where predicted probabilities are pushed to extreme values (0 or 1). The authors propose "gradient vector field surgery," a novel intervention that modifies the loss's partial derivatives to scale linearly with the prediction error. This approach maintains the high segmentation accuracy of region-based losses while significantly improving model calibration across 2D and 3D medical tasks.

**Strengths:**

* It is interesting to see the calibration effects could also be controlled through region-based loss functions like DICE as most of the earlier works involve handing the over-confidence issues in discriminative loss functions like Cross Entropy.
* The observations and intuitions presented by the authors regarding the gradient dynamics contributing to DICE loss looks feasible as I could very well relate it to the evolution of logit distribution in Cross Entropy and prediction based losses like Focal.
* The method was tested on diverse datasets including INbreast (mammography), FIVES (fundus images), BraTS-METS (brain metastasis), and KiTS (kidney tumors).

**Weaknesses:**

* How was the calibration metric calculated, did you consider only the foreground from target / foreground from prediction / union of foreground from both target and foreground ? https://www.sciencedirect.com/science/article/abs/pii/S1361841525000490 -- The ECE values are very low meaning, higher number of background pixels might have been included imbalacing the numbers.
* I understand the use of surgery specific to region-based losses, it would be very useful to the community if you can compare it with the calibration specific loss functions for segmentation proposed earlier including SVLS (https://arxiv.org/abs/2104.05788), NACL (https://arxiv.org/abs/2401.14487), and BWCR (https://arxiv.org/abs/2307.08163). That way, we can further appreciate the improvements.
* I would really appreciate it if you can add a discussion on how your method would behave in each of the different datasets. For instance, FIVES involves tubular structures, while INbreast expects blob-like segmentations. Besides, does your method have difficulty in handling multi-class over single-class ?

**Detailed Comments:**

Please refer weakness section

**Justification Of Final Rating:**

Appreciate the authors for addressing the comments. I like dice being used as the base for improving calibration. The proposed solution was also mathematically presented and rigorously evaluated. AI in medical imaging is already moving to production, hence calibration is key.

**Justification Of The Preliminary Rating:**

Calibration is an important area of research to move the algorithms to production. While calibration have been studied in cross entropy, it is not yet explored in other region-specific loss functions like DICE.

**Questions To Address In The Rebuttal:**

Please refer weakness section

---

> ### Author Response · Authors · 2026-01-25
>
> We thank you for your interest in our work and your questions regarding (1) the computation of our calibration metrics, (2) the comparison to other related methods, and (3) the applicability to different foreground shapes/structures and multi-class settings.
>
> **R wV9J:** On which pixels were the calibration metrics computed?
>
> **TLDR Answer:** The reported results are calculated on all pixels. A reevaluation using only the pixels in the union of the label and prediction foregrounds showed the same trends as reported in the paper, but with higher absolute calibration errors.
>
> We calculate the ECE value (and all other calibration metrics) on both foreground and background pixels. We believe this is a rational choice, especially since the calibration metrics fail to capture overconfident oversegmentation (i.e., all predictions for target foreground are 1.0) when only using target foreground. Similarly, these metrics fail to capture overconfident undersegmentation (i.e., all predictions for target foreground are 0.0 or 1.0, and all predictions for target background are 0.0) when using only the predicted foreground. Calculating the ECE on the "active" regions defined by the union of binarized predicted foreground and target foreground \cite{murugesan2025neighbor} is a compelling alternative to standard global evaluation. We reevaluated the calibration metrics on the union to compare them with our current evaluation and present the results in Appendix H and in the table below. As you mentioned, ECE values are substantially larger when the evaluation is limited to the "active" region. However, the observed improvements in calibration remain unchanged, not only for ECE but also for other metrics, including MCE, NLL, and Brier score. Notably, NACL demonstrates stronger calibration on active regions compared to the all-pixel evaluation on the KiTS dataset. We hypothesize that this stems from NACL's logit constraint, which enforces a universal reduction in confidence. While this effect is beneficial for preventing overconfidence in foreground regions, it suppresses confidence across the extensive background areas. Due to the high class imbalance in KiTS, this background effect dominates the global calculation, leading to larger NLL and ECE values compared to the active region evaluation.
>
> | Dataset | Method | NLL ↓ | ECE ↓ | MCE ↓ | Brier ↓ | Dice (%) ↑ |
> | :--- | :--- | :---: | :---: | :---: | :---: | :---: |
> | **INbreast (Active)** | Cross Entropy | 1.1372 ± 0.1571 | 0.3229 ± 0.0502 | 0.5940 ± 0.2130 | 0.3085 ± 0.0290 | 66.20 ± 3.57 |
> | | NACL | 1.2444 ± 0.2337 | 0.3167 ± 0.0427 | 0.6731 ± 0.2730 | 0.3064 ± 0.0347 | 68.22 ± 3.57 |
> | | NACL + Dice | 2.0573 ± 0.5311 | 0.3367 ± 0.0332 | 0.5938 ± 0.0380 | 0.3321 ± 0.0297 | 72.07 ± 2.57 |
> | | SVLS | 1.1109 ± 0.1555 | 0.3089 ± 0.0295 | 0.6844 ± 0.2495 | 0.2908 ± 0.0233 | 68.35 ± 2.90 |
> | | SVLS + Dice | 1.7828 ± 0.3316 | 0.3535 ± 0.0367 | 0.5863 ± 0.0428 | 0.3470 ± 0.0304 | 68.76 ± 2.70 |
> | | Dice++ | 1.2198 ± 0.0628 | 0.3207 ± 0.0251 | 0.5231 ± 0.0293 | 0.3137 ± 0.0166 | 67.51 ± 3.17 |
> | | CE + Dice | 2.1938 ± 0.3776 | 0.3411 ± 0.0270 | 0.5930 ± 0.0353 | 0.3355 ± 0.0256 | 71.76 ± 2.10 |
> | | CE + Dice - Surgery | **0.8807*** ± 0.0926 | _0.2427*_ ± 0.0322 | _0.4283*_ ± 0.0388 | **0.2470*** ± 0.0250 | **74.39** ± 3.34 |
> | | Dice | 3.2670 ± 1.3015 | 0.3996 ± 0.0433 | 0.6189 ± 0.0613 | 0.3940 ± 0.0401 | 65.06 ± 3.31 |
> | | Dice - Surgery | _0.9670_ ± 0.1173 | **0.2397*** ± 0.0313 | **0.4228*** ± 0.0548 | _0.2496*_ ± 0.0257 | _74.34*_ ± 2.77 |
> | | Tversky | 3.6811 ± 0.6354 | 0.3981 ± 0.0238 | 0.6365 ± 0.0217 | 0.3912 ± 0.0213 | 67.63 ± 2.89 |
> | | Tversky - Surgery | 1.0767* ± 0.1626 | 0.2842* ± 0.0405 | 0.4754* ± 0.0527 | 0.2808* ± 0.0307 | 69.34 ± 3.77 |
> | **FIVES (Active)** | Cross Entropy | 0.5448 ± 0.0121 | 0.1265 ± 0.0017 | 0.2856 ± 0.0088 | 0.1465 ± 0.0017 | 87.54 ± 0.22 |
> | | NACL | 0.5324 ± 0.0071 | _0.1253_ ± 0.0021 | _0.2844_ ± 0.0109 | 0.1450 ± 0.0012 | 87.60 ± 0.14 |
> | | NACL + Dice | 0.7641 ± 0.0341 | 0.1597 ± 0.0033 | 0.4003 ± 0.0106 | 0.1634 ± 0.0028 | 87.94 ± 0.11 |
> | | Dice++ | 0.5365 ± 0.0039 | 0.1343 ± 0.0014 | 0.3286 ± 0.0127 | 0.1452 ± 0.0010 | 87.97 ± 0.07 |
> | | ACE | 0.8581 ± 0.0274 | 0.1514 ± 0.0044 | 0.2927 ± 0.0116 | 0.1688 ± 0.0052 | 85.96 ± 0.45 |
> | | CE + Dice | 0.7682 ± 0.0160 | 0.1594 ± 0.0025 | 0.3972 ± 0.0106 | 0.1634 ± 0.0020 | _87.99_ ± 0.16 |
> | | CE + Dice - Surgery | 0.5340* ± 0.0264 | **0.1249*** ± 0.0050 | **0.2815*** ± 0.0110 | _0.1447*_ ± 0.0027 | 87.70 ± 0.09 |
> | | Dice | 2.7751 ± 0.3224 | 0.1947 ± 0.0050 | 0.5847 ± 0.0264 | 0.1940 ± 0.0049 | **88.03** ± 0.25 |
> | | Dice - Surgery | _0.5271*_ ± 0.0247 | 0.1283* ± 0.0050 | 0.2999* ± 0.0131 | 0.1451* ± 0.0024 | 87.60 ± 0.12 |
> | | Tversky | 2.5810 ± 0.2671 | 0.1969 ± 0.0018 | 0.5727 ± 0.0081 | 0.1960 ± 0.0019 | 87.77 ± 0.21 |
> | | Tversky - Surgery | **0.5189*** ± 0.0155 | 0.1258* ± 0.0029 | 0.2953* ± 0.0150 | **0.1432*** ± 0.0021 | 87.79 ± 0.25 |

---

> ### Author Response · Authors · 2026-01-25
>
> **Active Region Analaysis for the KiTS dataset**
>
> | | Method | NLL ↓ | ECE ↓ | MCE ↓ | Brier ↓ | DSC (%) ↑ |
> | :--- | :--- | :---: | :---: | :---: | :---: | :---: |
> | **KiTS Tumor** | CE | 1.6627 | 0.3306 | 0.5117 | 0.6870 | 64.79 |
> | | SVLS | **0.9213** | 0.2615 | 0.4867 | 0.5249 | 70.68 |
> | | SVLS + Dice | 2.5133 | 0.2994 | 0.5438 | 0.5969 | 75.82 |
> | | NACL | _0.9398_ | 0.3033 | 0.4730 | 0.6263 | 64.48 |
> | | NACL + Dice | 1.3766 | 0.2981 | 0.5567 | 0.5918 | 74.56 |
> | | Dice++ | 6.3755 | _0.2580_ | 0.4445 | 0.5244 | 75.57 |
> | | CE + Dice | 8.9081 | 0.3042 | 0.5664 | 0.6041 | 75.77 |
> | | CE + Dice - Surgery | *1.2973* | **0.2344** | **0.4132** | **0.4830** | **76.75** |
> | | Dice | 5.5300 | 0.3148 | 0.6309 | 0.6253 | _*76.62*_ |
> | | Dice - Surgery | *1.2977* | *0.2611* | _*0.4393*_ | _*0.5193*_ | 74.01 |
> | | Tversky | 7.3410 | 0.3506 | 0.6110 | 0.6961 | 73.10 |
> | | Tversky - Surgery | *1.5339* | *0.2671* | *0.4723* | *0.5258* | *73.87* |

---

> ### Author Response · Authors · 2026-01-25
>
> **R wV9J:** It would be very useful to the community if you can compare it with the calibration specific loss functions for segmentation ([1, 2, 3]).
>
> **TLDR Answer:** We now conceptually and empirically compare additional methods to our proposed solution. We have added the respective sections to the related work and results sections of the paper and summarized our findings below.
>
> Thank you for referencing the interesting related literature that is concerned with the regularization of logit magnitudes and solutions for boundary-annotation inaccuracies inherent to the medical-image annotation process to improve model calibration [1,2,3]. Spatially varying labels smoothing (SVLS) [1] takes inspiration from label smoothing, but smooths labels not consistently across all pixels, but specifically for those with varying neighbor annotations (i.e., boundary pixels). This method improves calibration for brain tumor, kidney tumor, long nodule, and prostate zone segmentation. Neighbor-Aware Calibration (NACL) [2] reformulates and extends SVLC by treating it as an additional penalty term rather than explicitly framing it as label smoothing, efficiently combining logit constraints with a specific focus on boundary annotation uncertainty. This allows flexible weighting of the initial optimization objective, along with an additional boundary-uncertainty aware logit constraint. Finally, boundary-weighted consistency regularization (BWCR) forces logit consistency across corresponding pixels from different augmented versions of the same input. All these approaches improve calibration by accounting for the inherent error susceptibility at annotation boundaries and the positive effect of limited logit magnitudes and differences on calibration [4]. This is different from our work, which specifically targets the pathological overconfidence of region-based losses in stand-alone or compound usage. Inspired by the logit analysis shown in [2] (Figure 4), we performed an extensive analysis of the logit magnitudes for our method and found that our surgery on the partials results in a substantial reduction in logit magnitudes compared to standard region-based losses. We added the quantitative results of these analyses in Appendix K of the updated paper.
>
>
> | Dataset | Method | Target FG (FG Logit) | Target FG (BG Logit) | Target BG (FG Logit) | Target BG (BG Logit) | Dice (%) ↑ |
> | :--- | :--- | :---: | :---: | :---: | :---: | :---: |
> | **INbreast** | CE + Dice | 3.51 ± 2.23 | -1.07 ± 1.27 | -5.39 ± 0.76 | 6.31 ± 0.71 | 71.76 ± 2.10 |
> | | CE + Dice + Tunable Grad | 1.52 ± 0.90 | -1.33 ± 0.89 | -4.31 ± 0.53 | 4.93 ± 0.47 | 74.39 ± 3.34 |
> | | Dice | 5.18 ± 2.34 | -1.42 ± 3.43 | -5.26 ± 1.08 | 6.18 ± 0.81 | 65.06 ± 3.31 |
> | | Dice Tunable Grad | 1.87 ± 1.89 | -0.67 ± 1.48 | -4.41 ± 0.40 | 5.53 ± 0.65 | 74.34 ± 2.77 |
> | | Tversky | 5.97 ± 0.97 | -1.79 ± 2.55 | -5.39 ± 0.69 | 6.34 ± 0.38 | 67.63 ± 2.89 |
> | | Tversky Tunable Grad | 2.28 ± 1.67 | -0.33 ± 1.76 | -4.52 ± 0.81 | 5.58 ± 0.54 | 69.34 ± 3.77 |
> | **FIVES** | CE + Dice | 3.71 ± 1.13 | -4.18 ± 1.07 | -4.08 ± 0.06 | 4.57 ± 0.14 | 87.99 ± 0.16 |
> | | CE + Dice + Tunable Grad | 2.71 ± 1.03 | -2.51 ± 1.02 | -3.19 ± 0.24 | 3.71 ± 0.22 | 87.70 ± 0.09 |
> | | Dice | 26.42 ± 13.03 | -29.12 ± 18.00 | -7.80 ± 1.18 | 7.74 ± 0.59 | 88.03 ± 0.25 |
> | | Dice Tunable Grad | 3.66 ± 0.18 | -1.91 ± 0.45 | -2.92 ± 0.28 | 3.52 ± 0.19 | 87.60 ± 0.12 |
> | | Tversky | 18.79 ± 11.26 | -18.51 ± 12.57 | -6.82 ± 1.12 | 7.03 ± 0.79 | 87.77 ± 0.21 |
> | | Tversky Tunable Grad | 3.40 ± 0.99 | -2.36 ± 0.94 | -2.96 ± 0.12 | 3.49 ± 0.16 | 87.79 ± 0.25 |

---

> ### Author Response · Authors · 2026-01-25
>
> **R wV9J:** Can our method handle multi-class segmentation tasks?
>
> **TLDR Answer:** Yes. We outline the multiclass extension below.
>
> An extension to the multiclass setting is possible. For the standard multiclass generalization, treating the problem as a combination of one-versus-rest problems, we can again define vector fields and perform "descent" on them. For scalar losses, the resulting gradient field is the sum of the gradient fields from the individual loss components. E.g. for 3 classes, $L_{MC} = L_{C1} + L_{C2}+ L_{C3}$ and $\nabla_\textbf{z}L_{MC} = \nabla_\textbf{z}L_{C1} + \nabla_\textbf{z}L_{C2} + \nabla_\textbf{z}L_{C3}$. Using our approach, the combined optimization objective would also be the sum of the vector fields defined on the individual optimization components: $\\mathcal{F}\_{MC}(\\mathbf{z}) = \\mathcal{F}\_{C1}(\\mathbf{z}) + \\mathcal{F}\_{C2}(\\mathbf{z}) + \\mathcal{F}\_{C3}(\\mathbf{z}) $.
>
> Importantly, we define our vector fields at the logit level, which means we need to account for the softmax's coupling of logits in the multiclass setting. In our implementation, shown in Appendix D, we already do this because we model the binary segmentation problem with background and foreground logits.
>
> **R wV9J:** How does our approach behave in different datasets with different foreground shapes?
>
> **TLDR Answer:** Our method performs best on datasets with high class imbalance or region size variations. We do not observe clear differences in our method's behavior depending on foreground shapes. We ran additional evaluations to test whether our method behaves differently for specific foreground sizes.
>
> We do not see a specific advantage/disadvantage for any particular geometric shape of foreground structures. However, we expect class ratios to affect the utility of our method. Given that our method includes region-loss-specific adaptive handling of region/class-size imbalance, it performs best in cases where region-based losses are effective. This is the case in scenarios of (1) extreme class imbalance [5,6,7], e.g., in the KiTs Tumor and BraTs Metastasis segmentation tasks, and in scenarios of (2) strong region size variation, e.g., in the INBreast dataset, where mass sizes vary drastically. In cases of relatively large foreground fractions combined with stable foreground-to-background ratios, we do not expect our method to outperform the standard cross-entropy loss. This is exactly what we observe in the FIVES dataset. Aside from this, we do not observe differences in our method's behavior that depend on the target structures' shapes (e.g., blob-like or tubular).
>
> | Lesion Size | Metric | Dice++ | CE + Dice | CE + Dice Surg. | Dice | Dice Surg. | Tversky | Tversky Surg. |
> | :--- | :--- | :--- | :--- | :--- | :--- | :--- | :--- | :--- |
> | **Small** | Det. | 0.9221 | 0.8961 | 0.9481 | 0.9091 | 0.9091 | 0.8961 | 0.9481 |
> | | ECE | 0.2272 | 0.3112 | 0.2172 | 0.2764 | 0.2182 | 0.3586 | 0.1664 |
> | **Large** | Det. | 0.9872 | 0.9615 | 0.9359 | 0.9615 | 0.9744 | 0.9615 | 0.9487 |
> | | ECE | 0.0976 | 0.1607 | 0.1279 | 0.1611 | 0.0991 | 0.2128 | 0.1170 |
> | **Very Large** | Det. | 1.000 | 0.9744 | 0.9871 | 0.9872 | 0.9872 | 1.000 | 0.9744 |
> | | ECE | 0.055 | 0.0949 | 0.0610 | 0.0949 | 0.0589 | 0.1134 | 0.0558 |

---

> ### Author Response · Authors · 2026-01-25
>
> **References**
>
> [1] Islam, M., & Glocker, B. (2021, June). Spatially varying label smoothing: Capturing uncertainty from expert annotations. In international conference on information processing in medical imaging (pp. 677-688). Cham: Springer International Publishing.
>
> [2] Murugesan, B., Vasudeva, S. A., Liu, B., Lombaert, H., Ayed, I. B., & Dolz, J. (2025). Neighbor-aware calibration of segmentation networks with penalty-based constraints. Medical Image Analysis, 101, 103501.
>
> [3] Karani, N., Dey, N., & Golland, P. (2023, October). Boundary-weighted logit consistency improves calibration of segmentation networks. In International conference on medical image computing and computer-assisted intervention (pp. 367-377). Cham: Springer Nature Switzerland.
>
> [4] Müller, R., Kornblith, S., & Hinton, G. E. (2019). When does label smoothing help?. Advances in neural information processing systems, 32.
>
> [5] Eelbode, T., Bertels, J., Berman, M., Vandermeulen, D., Maes, F., Bisschops, R., & Blaschko, M. B. (2020). Optimization for medical image segmentation: theory and practice when evaluating with dice score or jaccard index. IEEE transactions on medical imaging, 39(11), 3679-3690.
>
> [6] Ma, J., Chen, J., Ng, M., Huang, R., Li, Y., Li, C., ... & Martel, A. L. (2021). Loss odyssey in medical image segmentation. Medical image analysis, 71, 102035.
>
> [7] Sudre, C. H., Li, W., Vercauteren, T., Ourselin, S., & Jorge Cardoso, M. (2017, September). Generalised dice overlap as a deep learning loss function for highly unbalanced segmentations. In International Workshop on Deep Learning in Medical Image Analysis (pp. 240-248). Cham: Springer International Publishing.

---

### Official Review · Reviewer_jZg4 · 2026-01-10

**Confidence:** 4
**Preliminary Rating:** 4
**Final Rating:** 5

**Summary:**

The authors address the overconfidence of region-based losses like Dice by proposing "gradient vector field surgery". Instead of using a scalar loss, they directly define a gradient vector field that scales linearly with prediction error while retaining Dice's robustness to class imbalance. Experiments on 2D and 3D datasets show this approach significantly improves calibration metrics without compromising segmentation accuracy.

**Strengths:**

* The underlying problem of loss miscalibration and the proposed methodology are presented with good clarity. The authors make the novel concept of gradient vector field surgery intuitive and easy to understand.
* The paper addresses a very interesting and clinically relevant topic regarding the dice loss in medical image segmentation. The empirical results demonstrate consistent performance improvements over established baselines across multiple 2D and 3D datasets.

**Weaknesses:**

The Dice similarity coefficient is a clinically intuitive metric for evaluating segmentation. As observed in Tables 1 and 2, using gradient vector field surgery does not consistently yield higher Dice scores, sometimes resulting in decreases compared to the baselines.

**Detailed Comments:**

The standard deviation provided in table 1 but not in table 2.

**Justification Of Final Rating:**

I would like to thank the authors for thoroughly addressing my concern regarding the clinical applicability of the proposed method and the scenario in which it demonstrates improvement, as clarified in the manuscript. I am happy to increase the final rating accordingly.

**Justification Of The Preliminary Rating:**

This paper makes a significant contribution by proposing a novel "gradient vector field surgery" that resolves the pathological overconfidence of the Dice loss while retaining its robustness to class imbalance.

**Questions To Address In The Rebuttal:**

Discuss which types of lesions will benefit from the proposed loss function.

---

> ### Author Response · Authors · 2026-01-25
>
> We thank you for the feedback and address your questions below. We hope that we can explain the interpretation of our results further and provide additional ablation studies regarding different lesion sizes.
>
> **R jZg4:** The proposed method does not consistently improve Dice scores.
>
> **TLDR Answer:** That is correct. The achieved dice scores are generally not statistically significantly different from the baselines. Our method significantly improves model calibration while maintaining Dice accuracy.
>
> Our method is designed to theoretically explain and address the known overconfidence problem in region-based losses. In our empirical evaluation, we observe across diverse 2D and 3D tasks that the overconfidence problem is largely resolved, with drastic improvements in multiple calibration metrics, including ECE, MCE, NLL, and Brier score. We now also include statistical tests for the 2D datasets, showing that in addition to the large effect sizes, the performance improvements are also statistically significant. Dice score is, as you note, an important metric for clinical evaluation, especially in tasks where volumetric information is important; however, it also has known limitations, e.g., in lesion segmentation for metastasis, where lesion detection (also of small lesions) is often more important than perfect volumetric accuracy. Calibrated model outputs can play an important role in these cases, where clinicians can focus on the verification of areas where models are uncertain, if lesions are present or not (see Figure 3). Finally, our results do not provide conclusive evidence that our method improves dice scores, nor do they show that dice scores suffer from the method. On the other hand, the evidence for improved calibration of model outputs is conclusive in effect size and significance.
>
> **R jZg4:** Discuss which types of lesions will benefit from the proposed loss function.
>
> **TLDR Answer:** Our method performs best in datasets exhibiting large region/class size imbalances. We did not observe conclusive evidence that our method's performance differs between lesion types (brain, kidney, breast) or between lesion sizes within a foreground/background-imbalanced dataset.
>
> Given that our method includes region-loss-specific adaptive handling of region/class-size imbalance, it performs best in cases where region-based losses are effective. This is the case in scenarios of (1) extreme class imbalance [1,2,3], e.g., in the KiTs Tumor and BraTs Metastasis segmentation tasks, and in scenarios of (2) strong region size variation, e.g., in the INBreast dataset, where mass sizes vary drastically. In cases of relatively large foreground fractions combined with stable foreground-to-background ratios, we do not expect our method to outperform the standard cross-entropy loss. This is exactly what we observe in the FIVES dataset.
>
> To explore whether the benefits of our method depend on lesion size, we ran an additional ablation study on the KiTS dataset, measuring detection performance and calibration (via ECE) per lesion (i.e., on the lesions' foreground voxels in the label). We then aggregated the results for different lesion sizes. We added the results to the paper in Appendix I and in the table below. We observe consistent calibration improvements when applying our proposed gradient field surgery across all tumor sizes. Furthermore, the results indicate that calibration improvements are slightly more pronounced for small tumors. %We hypothesize that improved calibration is most beneficial for small tumors, where uncertainty is naturally higher. In these borderline cases, accurate probability estimates are critical for successful detection because they help push the predicted probabilities of these difficult cases close to the detection threshold, thereby improving detection.
> Notably for the KiTS dataset, the foreground/background ratio remains highly imbalanced even for samples with large and very large lesions, making region-based loss functions effective.
>
> | Lesion Size | Metric | Dice++ | CE + Dice | CE + Dice Surg. | Dice | Dice Surg. | Tversky | Tversky Surg. |
> | :--- | :--- | :--- | :--- | :--- | :--- | :--- | :--- | :--- |
> | **Small** | Det. | 0.9221 | 0.8961 | 0.9481 | 0.9091 | 0.9091 | 0.8961 | 0.9481 |
> | | ECE | 0.2272 | 0.3112 | 0.2172 | 0.2764 | 0.2182 | 0.3586 | 0.1664 |
> | **Large** | Det. | 0.9872 | 0.9615 | 0.9359 | 0.9615 | 0.9744 | 0.9615 | 0.9487 |
> | | ECE | 0.0976 | 0.1607 | 0.1279 | 0.1611 | 0.0991 | 0.2128 | 0.1170 |
> | **Very Large** | Det. | 1.000 | 0.9744 | 0.9871 | 0.9872 | 0.9872 | 1.000 | 0.9744 |
> | | ECE | 0.055 | 0.0949 | 0.0610 | 0.0949 | 0.0589 | 0.1134 | 0.0558 |

---

> ### Author Response · Authors · 2026-01-25
>
> **References**
>
> [1] Eelbode, T., Bertels, J., Berman, M., Vandermeulen, D., Maes, F., Bisschops, R., & Blaschko, M. B. (2020). Optimization for medical image segmentation: theory and practice when evaluating with dice score or jaccard index. IEEE transactions on medical imaging, 39(11), 3679-3690.
>
> [2] Ma, J., Chen, J., Ng, M., Huang, R., Li, Y., Li, C., ... & Martel, A. L. (2021). Loss odyssey in medical image segmentation. Medical image analysis, 71, 102035.
>
> [3] Sudre, C. H., Li, W., Vercauteren, T., Ourselin, S., & Jorge Cardoso, M. (2017, September). Generalised dice overlap as a deep learning loss function for highly unbalanced segmentations. In International Workshop on Deep Learning in Medical Image Analysis (pp. 240-248). Cham: Springer International Publishing.

---

### Author Rebuttal · Authors · 2026-01-25

**Rebuttal:**

Dear Reviewers,

We would like to thank you all for your insightful and constructive feedback. We have addressed each comment in the individual responses below and revised the manuscript accordingly (changes are in violet). We will try to answer any remaining or additional questions in the discussion phase as soon as possible.

- **Comparison to Calibration Methods**: We expanded our related work and results to include conceptual and empirical comparisons with calibration-specific loss functions and added a logit magnitude analysis (Sections 2 and 4, Appendix K).
- **Active Region Evaluation**: In addition to our evaluation on all pixels, we provide a re-evaluation of calibration metrics on the union of predicted and ground truth foregrounds (Appendix H).
- **Analysis of effectiveness on different foreground shapes/sizes/structures**: We discuss in which cases our method is most effective and provide additional empirical results for different lesion sizes (Appendix I).
- **Experiments on other architectures**: We provide additional experiments demonstrating the applicability of our gradient field surgery to Transformer-based architectures (Appendix J).
- **Extension to multi-class settings**: We discuss how our method can be applied to multi-class segmentation tasks.

We look forward to engaging further during the discussion phase to answer any remaining questions.

Many thanks and best wishes,

the authors

**Supporting Material:**

/attachment/c795fc49d7ad6890f7c4915beba8cfd7a250999f.pdf

---

### Author Response · Authors · 2026-01-29
**Discussion period**

Dear reviewers,

Thank you once again for your feedback on our work. You raised many important questions, which strengthened the contribution of our work. Following your suggestions, we have made a considerable effort to consolidate the analysis of the proposed method. We hope that our response addressed your concerns, and we look forward to receiving your further feedback during the discussion phase.

Many thanks and best wishes,

the authors

---

### Meta-Review · Area_Chair_sVpg · 2026-02-07

**Recommendation:** Accept (Oral)
**Confidence:** 5

**Metareview:**

The paper proposes an analysis, explanation, and solution to the miscalibration that can be encoutered in segmentation when using Dice loss (notably) during the training.

The work is valuable and timely, and all reviewers agree on the value and quality of the work.

I am on the fence between recommending an oral, or a poster. But that is something that the program chairs will have to decide :) In any case, I'll happy to drop at the poster session for discussion with the authors.

Last, but not least: sharing at least some reference implementation of the work would be great: https://www.midl.io/reproducibility

---

### Decision · Program_Chairs · 2026-02-13

Accept (Poster)